# Profiling the bloodstream form and procyclic form *Trypanosoma brucei* cell cycle using single-cell transcriptomics

**Emma M Briggs[1,2]\*, Catarina A Marques[2], Guy R Oldrieve[1], Jihua Hu[1], Thomas D Otto[2], Keith R Matthews[1]**

[1]Institute for Immunology and Infection Research, School of Biological Sciences, University of Edinburgh, Edinburgh, United Kingdom; [2]Wellcome Centre for Integrative Parasitology, School of Infection & Immunity, University of Glasgow, Glasgow, United Kingdom

**Abstract** African trypanosomes proliferate as bloodstream forms (BSFs) and procyclic forms in the mammal and tsetse fly midgut, respectively. This allows them to colonise the host environment upon infection and ensure life cycle progression. Yet, understanding of the mechanisms that regulate and drive the cell replication cycle of these forms is limited. Using single-cell transcriptomics on unsynchronised cell populations, we have obtained high resolution cell cycle regulated (CCR) transcriptomes of both procyclic and slender BSF *Trypanosoma brucei* without prior cell sorting or synchronisation. Additionally, we describe an efficient freeze–thawing protocol that allows single-cell transcriptomic analysis of cryopreserved *T. brucei*. Computational reconstruction of the cell cycle using periodic pseudotime inference allowed the dynamic expression patterns of cycling genes to be profiled for both life cycle forms. Comparative analyses identify a core cycling transcriptome highly conserved between forms, as well as several genes where transcript levels dynamics are form specific. Comparing transcript expression patterns with protein abundance revealed that the majority of genes with periodic cycling transcript and protein levels exhibit a relative delay between peak transcript and protein expression. This work reveals novel detail of the CCR transcriptomes of both forms, which are available for further interrogation via an interactive webtool.

**\*For correspondence:**
emma.briggs@ed.ac.uk

**Competing interest:** The authors declare that no competing interests exist.

## Editor's evaluation

This important study maps changes in transcript levels over the cell cycle of two major developmental stages of the parasitic protist, Trypanosoma brucei. Single-cell RNA-seq on asynchronously replicating insect and mammlain-infective parasite stages identified over 1500 transcripts that are cell cycle regulated, significantly expanding the number of genes and cellular processes linked to cell cycle progression. Significantly, only some of these transcript levels are reflected in changes in corresponding protein levels, underlining the importance of both pre- and post-transcriptional regulatory processes in these divergent eukaryotes.

## Introduction

The *Trypanosoma brucei* life cycle involves developmental transitions between replicative and cell cycle arrested forms, the latter of which are primed for transmission between the mammal and tsetse fly, or vice versa (***Matthews, 2005***). Metacyclic trypomastigotes emerge from the replicating epimastigote population as arrested G0 forms in the tsetse fly salivary gland and express genes required for infection of the mammal, which occurs during their transfer during a bloodmeal (***Christiano et al., 2017***).

Metacyclics subsequently re-enter the cell cycle and differentiate into replicative slender bloodstream forms (BSFs). Slender BSFs proliferate and increase in parasitaemia before exiting the cell cycle via a quorum sensing mechanism (*Rojas and Matthews, 2019*; *Matthews, 2021*) and differentiating into G0 arrested stumpy BSFs, which express genes required for differentiation into replicating procyclic forms (PCFs) once in the tsetse fly midgut (*Silvester et al., 2017*).

The cell cycle of *T. brucei* broadly follows the typical eukaryotic progression through G1, S, G2, and M phases followed by cytokinesis. Although, trypanosomes are unusual in that the nuclear and mitochondrial genome is replicated and segregated prior to the nuclear genome in a precisely orchestrated sequence of events. While many canonical regulators remain unidentified, are absent, or have been replaced by trypanosomatid-specific factors. Several regulators have been identified (*Hammarton, 2007*; *Li, 2012*; *Wheeler et al., 2019*; *Passos et al., 2022*) including cdc2-related kinases (CRKs) (*Mottram and Smith, 1995*) and 13 cyclins (*Hammarton, 2007*; *Li, 2012*; *Wheeler et al., 2019*; *Passos et al., 2022*; *Lee and Li, 2021*), several of which have been linked to regulation of the *T. brucei* cell cycle phase transitions. Additionally, transcriptomic, proteomic, and phosphoproteomic analysis of semi-synchronised PCF populations have uncovered numerous cell cycle regulated (CCR) genes for further investigation (*Archer et al., 2011*; *Crozier et al., 2018*; *Benz and Urbaniak, 2019*). However, little overlap has been observed between these studies (*Benz and Urbaniak, 2019*), reflecting both the variation between experimental design and differences between transcript and protein regulation.

Single-cell transcriptomics (scRNA-seq) allows the transcriptomes of individual cells in a heterogenous, asynchronous population to be captured without the need to first isolate the target cell types by methods such as physical or chemical synchronisation and cell sorting. Continuous biological processes, such as cell cycle progression and cellular differentiation, can then be reconstructed in silico using trajectory inference and pseudotime approaches where cells are ordered by their progressive transcriptomic changes (*Tritschler et al., 2019*; *Wolfien et al., 2021*). Differential expression (DE) analysis across these ordered cells identifies genes with altered transcript levels during the process, and the dynamic change in transcript levels can be modelled.

scRNA-seq has been used effectively to compare transcriptomes of various *T. brucei* life cycle stage forms, including those extracted from tsetse flies to analyse the development of metacyclics in the salivary gland (*Hutchinson et al., 2021*; *Vigneron et al., 2020*; *Howick et al., 2022*) and to investigate slender to stumpy differentiation of BSFs in vitro (*Briggs et al., 2021a*). These studies mainly employed droplet-based methods (Drop-seq [*Hutchinson et al., 2021*] and Chromium 10× Genomics [*Vigneron et al., 2020*; *Briggs et al., 2021a*]) to recover higher cell numbers and relied on live, freshly derived parasites to ensure sufficient transcript recovery per cell (*Briggs et al., 2021b*). However, the need to use live parasites for droplet-based methods restricts usage of these approaches in experiments where high cell numbers or multiple time points are required, for example when modelling a developmental processes with trajectory inference methods (*Tritschler et al., 2019*). A previous attempt to use methanol fixed BSFs with Chromium technology yielded low transcript recovery per cell (*Briggs et al., 2021b*).

In this study, we profile the dynamic transcript changes during the cell cycle of laboratory cultured 'monomorphic' slender BSFs (refractory to stumpy differentiation and so quorum sensing dependent cell cycle exit) and PCFs. For each form, we also compare 10× Chromium generated transcriptomes from live parasites and parasites cryopreserved with glycerol in liquid nitrogen (LN$_2$). We find cryopreservation causes limited changes to the transcriptome of BSF and PCF *T. brucei*, highlighting cryopreservation as a valuable method of sample preservation for scRNA-seq analysis of trypanosomes. This will allow for future studies involving multiple conditions, or sampling over a time course using trypanosomes and, likely, other kinetoplastida and apicomplexan parasites. Periodic pseudotime inference was applied to the resulting data to model the cell cycle progression of both BSF and PCF *T. brucei*, allowing the genes with CCR transcripts to be identified in each form. Comparison with existing high-quality PCF proteomic datasets (*Crozier et al., 2018*; *Benz and Urbaniak, 2019*) further revealed a relative offset in peak transcript and protein levels for at least 50% of genes exhibiting CCR with respect to transcripts and proteins. Comparison between BSFs and PCFs identified genes with shared or life cycle stage-specific CCR transcripts, revealing both common and developmentally specific CCR factors, as well as apparent differences in the S–G2 transition between forms.

## Results

### Cryopreservation of *T. brucei* for Chromium single-cell transcriptomics

Generating scRNA-seq data with droplet-based technology Chromium (10× Genomics) currently requires live trypanosome samples in order to recover a high number of transcripts per cell (*Briggs et al., 2021b*). To test whether *T. brucei* could be stored prior to processing, we compared the impact of cryopreservation using 10% glycerol on live cell recovery when using a slow thawing protocol (*Figure 1—figure supplement 1*, methods). Using motility as a measure of parasite viability indicated both BSF and PCF cells recovered with high viability after freezing with 10% glycerol, with each form maintaining at least 90% cellular motility after 28 days of cryostorage (*Figure 1—figure supplement 1*). When returned to culture, parasites showed a delayed return to normal growth rates (*Figure 1—figure supplement 1*) indicating samples should be processed for scRNA-seq immediately after thawing to reflect their transcript status when cryopreserved.

Using this approach, replicating BSF and PCF *T. brucei* were processed for Chromium scRNA-seq 'fresh' from in vitro culture or after 13 days of storage with 10% glycerol in $LN_2$, hereafter referred to as 'frozen' (*Figure 1—figure supplement 2*). Frozen samples were thawed on day 13 and processed alongside the fresh samples taken directly from culture, thus fresh and frozen samples contain biological replicates and were subjected to scRNA-seq in the same batch. Cryopreservation had little effect on the raw data quality (*Figure 1*; *Supplementary file 1*) with the total numbers of unique transcript counts (unique molecular identifiers; UMIs) and features (encoding genes) detected per cell unaffected for either BSF or PCFs (*Figure 1A, B*). Additionally, the percentage of transcripts derived from the mitochondrial kDNA maxicircle genome was unchanged by the freezing and recovery procedures (*Figure 1C*). The percentage of kDNA-derived transcripts was higher in PCF compared to BSF, as expected: only PCFs require complexes III and IV for oxidative phosphorylation (*Smith et al., 2017*), components for which are encoded on the kDNA maxicircle (*Benne, 1985*). Higher average UMIs and features per cell were also recovered in PCF compared to BSF, in both fresh and frozen samples, although it is unclear if this is a biological phenomenon or if RNA extraction and capture is more efficient from PCFs. After filtering the transcriptomes based on these parameters to remove those of low quality or likely multiplets (*Figure 1*), 81.7% and 81.04% of fresh and frozen BSFs cells were retained in the data leaving 2767 and 1599 total cells, respectively. For PCFs, 76.82% and 72.60% of fresh and frozen cells were retained, leaving 3305 and 4335 cells, respectively. The differences in total number of cells are likely due to variation in loading and cell capture between samples.

Principle component analysis (PCA) highlighted far great variability between samples of different life cycle forms (93% of variance), compared to the preparation method (i.e. fresh or frozen, 5% variance) (*Figure 1D*). Average transcript counts across cells for each gene were significantly correlated between fresh and frozen samples in both BSF and PCF forms (Pearson's $R = 0.977$ and 0.985, respectively) (*Figure 1E, F*), with few genes (BSF: 0.80% of genes captured, PCF: 0.55%) showing greater than twofold difference (*Supplementary file 2*). DE analysis comparing single-cell transcriptomes of fresh and frozen samples using MAST (*Finak et al., 2015*) revealed 17 genes altered in BSF (14 upregulated in fresh, 3 in frozen) and 19 genes (13 in fresh and 6 in frozen) between PCFs (adjusted p-value <0.05, FC >1.5) (*Figure 1G, H*; *Supplementary file 2*). Only one gene, which putatively encodes fructose-bisphosphate aldolase class-I, was differentially expressed in both forms, with higher expression in fresh samples (*Figure 1I*). Notably, procyclin-associated genes (PAGs) 1–5 were all upregulated in frozen PCFs (*Figure 1H*).

As no large-scale transcriptomic changes in response to cryopreservation were observed, and DE genes did not include those linked to cell cycle regulation, fresh and frozen samples were integrated as replicate samples to analyse the cell cycle of PCF and BSF *T. brucei*.

### The CCR transcriptome of PCF *T. brucei*

PCF scRNA-seq data from fresh and frozen samples were integrated and dimensional reduction was performed. Transcriptomes were then plotted in low dimensional space as unifold manifold approximation and projection (UMAP; *McInnes et al., 2018*) plots, where cells are arranged by transcriptional similarities and differences (*Figure 2A*). Using cell cycle phase markers (*Supplementary file 1*) identified previously using bulk-RNA-seq (*Archer et al., 2011*), each cell was labelled by phase (*Figure 2B*). Grouping by phase was evident in both samples, with each population arranging in a logical order according to cell cycle progression. The proportion of cells in each phase was similar between samples

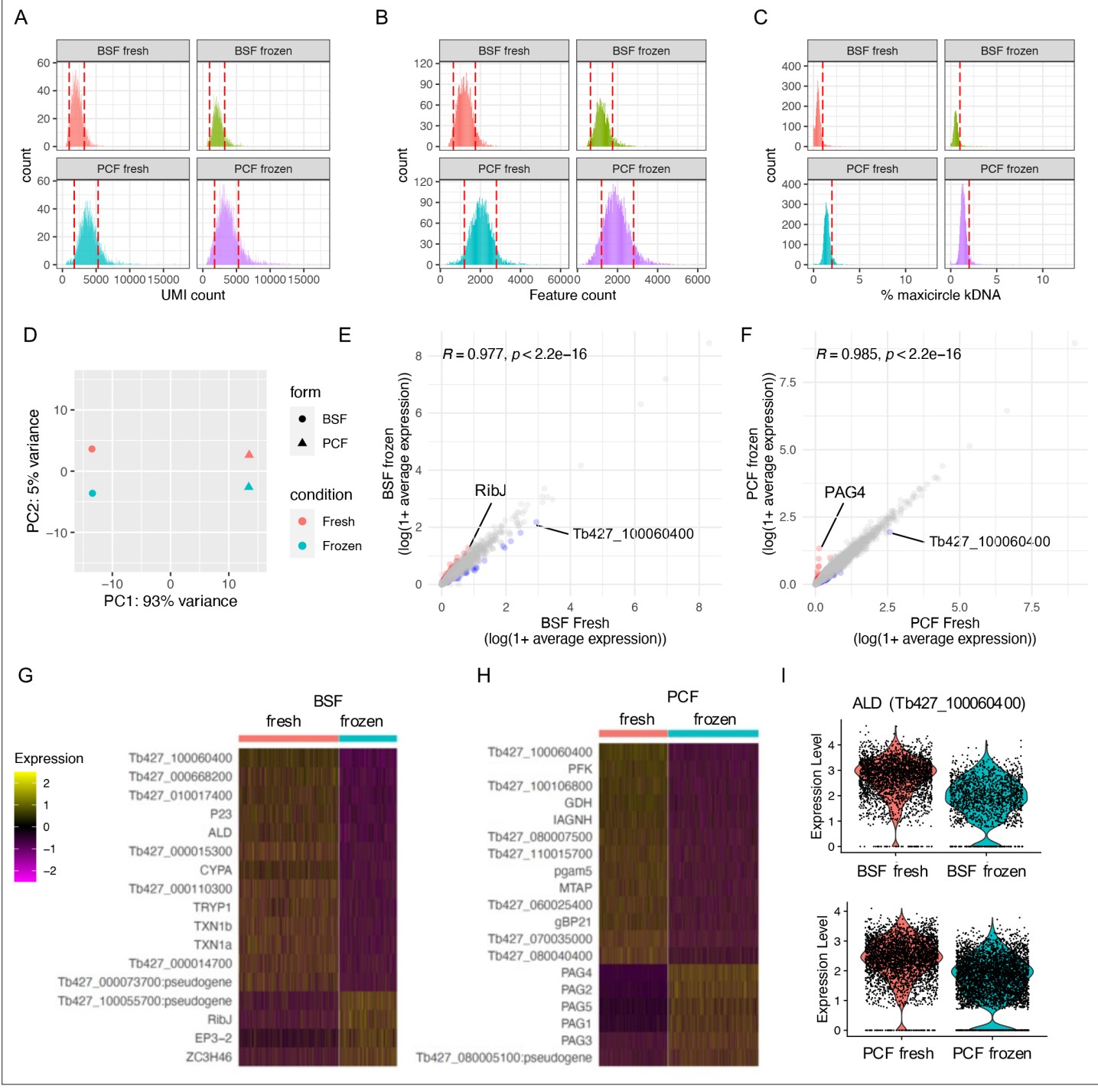

**Figure 1.** scRNA-seq of cryopreserved and fresh *T. brucei* bloodstream form (BSF) and procyclic form (PCF). (**A**) The unique molecular identifiers (UMI, *x*-axis) captured per cell (count, *y*-axis) by Chromium scRNA-seq with BSF and PCF taken fresh from in vitro culture (fresh) or after cryopreservation in liquid nitrogen ($LN_2$) (frozen). Red dashed lines indicate threshold used for QC filtering of each sample. (**B**) Number of genes (features, *x*-axis) for which transcripts were capture per cell. (**C**) Percentage of transcripts captured per cell that are encoded by genes on the mitochondrial maxicircle kDNA genome (% maxicircle kDNA, *y*-axis). (**D**) Top 2 components (PC1 and PC2) identified with PC analysis after pseudobulking all counts for each sample. Fresh (red) and frozen (blue) samples are shown for BSFs (circle) and PCFs (triangles). (**E**) Average expression of each gene across all cells for BSF fresh (*x*-axis) and BSF frozen (*y*-axis) plotted as log(1 + mean average count). Correlation coefficient and p-value of one-tailed Wilcox test is indicated above. Gene with increased fold change (FC) >2 in frozen sample are coloured red and those decreased in blue. (**F**) Average gene expression of PCF samples, as in E. (**G**) Scaled expression of genes DE between fresh and frozen BSF scRNA-seq (adjusted p-value <0.05, FC >1.5). Gene names are given when

*Figure 1 continued on next page*

*Figure 1 continued*

available, otherwise gene IDs are shown. (**H**) as in G for PCF samples. (**I**) Raw transcript counts (expression level) for fructose-bisphosphate aldolase (ALD; Tb427_100060400) in BSF (upper) and PCF (lower).

The online version of this article includes the following figure supplement(s) for figure 1:

**Figure supplement 1.** Effect of cryopreservation on *T. brucei* viability.

**Figure supplement 2.** Preparation of replicating bloodstream form (BSF) and procyclic form (PCF) *T. brucei* prior to scRNA-seq or cryopreservation.

---

(*Figure 2C*) and corresponded with the proportion of cells in G1 (1N), S (>1N <2N), and G2/M (2N) phases as assessed by flow cytometry analysis of DNA content (*Figure 2D*). Flow cytometry was performed prior to cryopreservation for frozen samples. A proportion of cells (fresh 6.23%, frozen 12.02%) did not elevate transcript levels of markers for any phase and so were named 'unlabelled' (grey; *Figure 2B, C*). The majority of unlabelled cells cluster with early G1 cells (*Figure 2B*). DE analysis between early G1 and unlabelled cells found 14 genes with adjusted p-value <0.05, yet none showed fold-change >1.5 (*Supplementary file 3*). These include three ribosomal proteins, a DEAD box helicase and a putative subunit of replicative protein A (RPA). It is possible that these cells are yet to re-enter the cell cycle and so do not over express any early G1 markers, or that the early G1 markers used here are insufficient to label all cells in this phase.

Pseudotime values were assigned using Cyclum, an autoencoder technique which projects cells on a nonlinear periodic trajectory (*Liang et al., 2020*). This is performed independently of the UMAP plotting and phase assignment described above. Cells ordered according to cell cycle progression and phases clearly separated in pseudotime, with the exception of early G1 and 'unlabelled' cells (*Figure 2E*). As expected, total RNA increased over pseudotime from early G1 (3221 median UMI per cell) to G2M (4077 median UMI) (*Figure 2—figure supplement 1*). Hence, DE analysis across pseudotime was performed using normalised counts to find CCR transcripts independent of total RNA increase. PseudotimeDE (*Song and Li, 2021*) was used to identify DE genes and thresholds for selecting significantly CCR genes were selected based on the detection of the previously identified CCR genes in PCF transcriptomic analysis (*Archer et al., 2011*; *Figure 2—figure supplement 2*). Using these cut-offs (false discovery rate [FDR] adjusted p-value <0.01, mean fold-change >1.5) 1550 significant CCR genes were identified (*Supplementary file 4*), including 399 of the 530 genes (75.28%) previously detected with the bulk RNA-seq approach (*Archer et al., 2011*; *Figure 2—figure supplement 3*, *Supplementary file 4*). Dynamic expression patterns were evident across the cell cycle (*Figure 2F*). Each gene was classified as peaking in a particular phase by comparing the average expression levels across cells for each phase. This revealed 77 (4.53%) genes with highest expression in early G1, 498 (29.31%) in late G1, 598 (35.20%) in S phase, and 526 (31.00%) in G2/M (*Supplementary file 4*, *Figure 2F*).

## Relative temporal relationship between RNA and protein levels in PCFs

To investigate the correlation between transcript and protein abundance during the PCF cell cycle, CCR genes defined by scRNA-seq above were compared with CCR proteins identified in two separate studies. *Crozier et al., 2018* and *Benz and Urbaniak, 2019* employed centrifugal elutriation to enrich for smaller G1 phase *T. brucei* PCFs which were then returned to culture and allowed to progress through the cell cycle in a semi-synchronised manner over time. Mass spectrometry was then employed to analyse protein samples taken as the cell population progressed through the cell cycle. Comparison of protein abundance in each sample then allowed CCR proteins to be identified. 427 and 370 genes (annotated in the WT427 2018 genome) were classified as encoding CCR proteins by Benz and Crozier, respectively, with 61 classed as CCR in both datasets. Of the 1550 genes with CCR transcripts in the present scRNA-seq data, 226 were classed as having CCR proteins in both or one of these studies (*Figure 2—figure supplement 3*). Proteomics analysis has lower sensitivity compared to transcriptomics, therefore not all scRNA-seq defined CCR genes are detected as proteins in these studies: 998 (64.39%) and 667 (43.03%) CCR genes were detected by Crozier and Benz, respectively. Of these, just 14.43% and 17.69% had been classified as CCR by Crozier (*Figure 2G*) and Benz (*Figure 2—figure supplement 4*), respectively. Thus, the majority of CCR transcripts do not result in CCR protein levels as defined by current methods. Proteins were not detected for 586 scRNA-seq

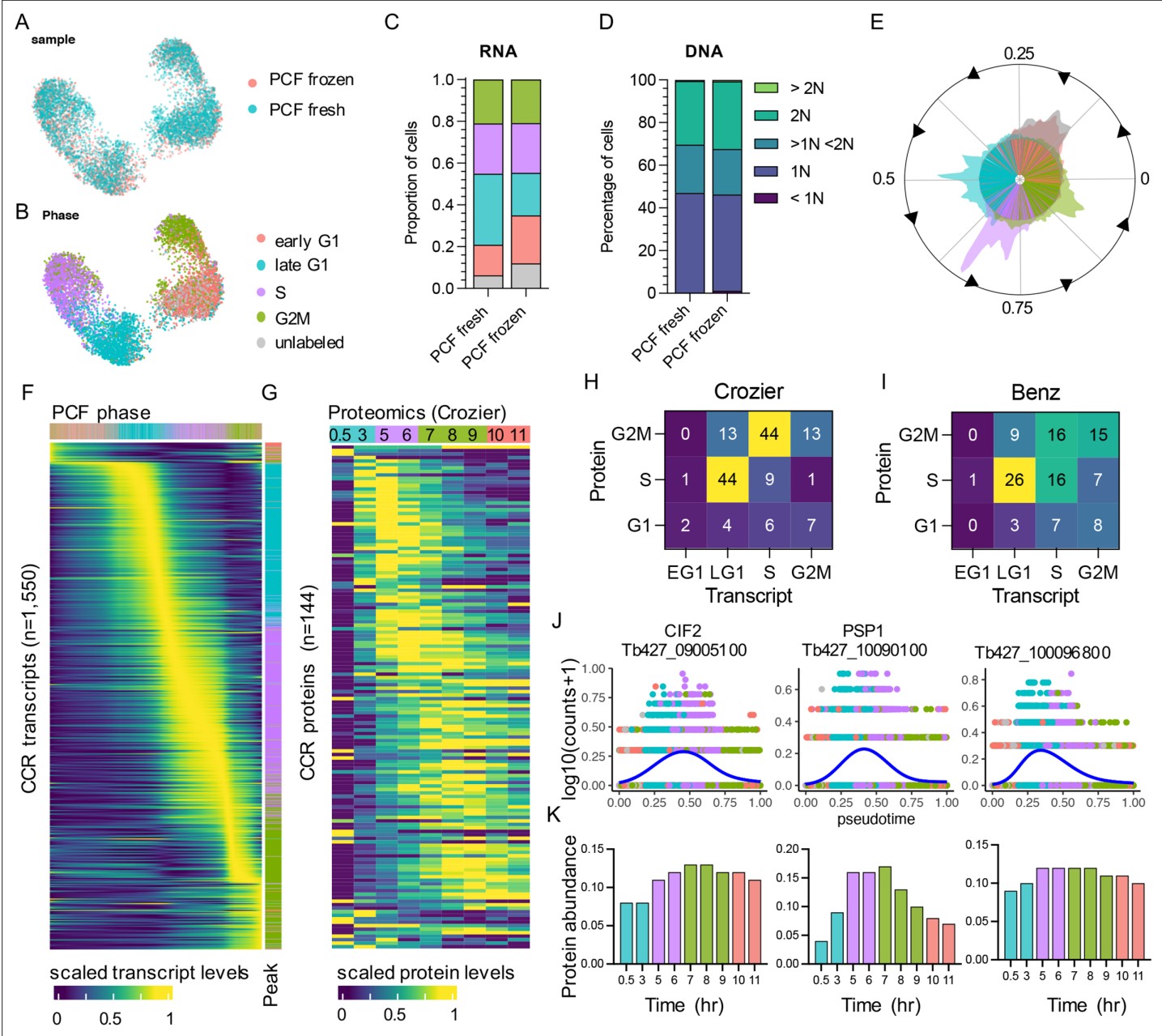

**Figure 2.** The cell cycle transcriptome of procyclic form (PCF) *T. brucei*. (**A**) Unifold manifold approximation and projection (UMAP) plot of integrated PCF transcriptomes from fresh (blue) and frozen (red) samples. (**B**) UMAP of PCF transcriptomes coloured by inferred cell cycle phase. (**C**) Proportion of cells assigned to each phase by transcriptomics as in B. Legend as in B. (**D**) Proportion of cells with DNA content assessed by flow cytometry. (**E**) Histogram of transcriptomes arranged in pseudotime (anti-clockwise) representing cell cycle progression. Each line in inner circle indicates one transcriptome coloured by phase as in B. Outer circle histogram of showing number of cells at each point in pseudotime (0–1). (**F**) Scaled transcript levels of cell cycle regulated (CCR) genes (rows), ordered by peak time, plotted across transcriptomes (columns) ordered in pseudotime. Top annotation indicates cell phase, right annotation indicates phase with highest expression of each gene. (**G**) Scaled protein abundance for 129 genes identified as CCR by Crozier et al., plotted in the same order as F. Time points are indicated in top annotation, coloured by the most enriched cell cycle phase for each sample. Numbers of genes with highest transcript expression in each phase analysed by scRNA-seq (*x*-axis) and highest protein level identified by Crozer et al. (**H**) and Benz et al. (**I**) proteomics studies. (**J**) Transcript counts of three genes (*y*-axis) plotted across pseudotime (*x*-axis). Each dot shows one transcriptome coloured by phase as in B. Blue line shows smoothed expression across pseudotime. (**K**) Protein abundance for the same genes as in J, previously identify as CCR by Crozier et al. Time point and colour of most enriched phase for each sample (*x*-axis) as in G.

The online version of this article includes the following figure supplement(s) for figure 2:

**Figure supplement 1.** Total RNA captured per cell across bloodstream form (BSF) and procyclic form (PCF) cell cycle progression.

*Figure 2 continued on next page*

*Figure 2 continued*

**Figure supplement 2.** Cell cycle regulated (CCR) gene selection thresholds.

**Figure supplement 3.** Comparison of cell cycle regulated (CCR) genes selected in transcriptomic and proteomic studies.

**Figure supplement 4.** Protein abundance levels across the procyclic form (PCF) cell cycle as defined by Benz et al.

CCR transcripts in either study, despite these not showing lower transcript abundance than those with detectable proteins, and so could not be compared (*Supplementary file 4*).

Plotting scaled CCR protein levels in the Crozier data (*Figure 2G*) and, to a lesser extent, in the Benz data (*Figure 2—figure supplement 4*) revealed dynamic abundance patterns across the cell cycle that broadly followed the dynamic transcript patterns identified by scRNA-seq. Comparing the relative timing of peak transcript and peak protein levels showed a common trend where transcript levels often peaked in the phase preceding the protein peak (*Figure 2H*). This broad pattern was observed for 66.67% and 47.22% of CCR genes when comparing transcripts to protein levels from Crozier (*Figure 2H*) and Benz studies (*Figure 2I*), respectively. Comparison to the Benz study indicated more genes peaking in the same phase for transcripts and proteins (37.78%), compared to the Crozier study (19.44%).

Just 24 genes were classified as CCR in scRNA-seq, bulk-RNA-seq (*Archer et al., 2011*), and both proteomic studies (*Figure 2—figure supplement 3*). These include genes with documented roles in the cell cycle: cyclin-dependent kinase CRK3 (Tb427_100054000), cyclin-dependent kinase regulatory subunit CKS1 (Tb427_110183500), cytokinesis initiation factors CIF1 (Tb427_110176500) and CIF2 (Tb427_090085100), and Cohesin subunit SCC3 (Tb427_100064300). Others include three homologues of *S. cerevisiae* Polymerase Suppressor PSP1 (Tb427_100090100, Tb427_110047000, and Tb427_110165900), and six genes encoding hypothetical proteins with no known function (Tb427_040026500, Tb427_040054600, Tb427_080009800, Tb427_100096800, Tb427_100120400, and Tb427_110082700). Transcript levels for these genes were raised (*Figure 2J*) prior to protein levels (*Figure 2K*, *Figure 2—figure supplement 4*).

## The CCR transcriptome of BSF *T. brucei*

The same approach was taken to analyse transcript dynamics during the BSF cell cycle. Transcriptomes from both the fresh and frozen samples (*Figure 3A*) arranged in low dimensional space according to phase, as assigned using bulk RNA-seq defined markers (*Figure 3B*). Notably, S and G2M BSF cells display less separation in UMAP plots compared (*Figure 3B*) to PCFs (*Figure 2B*), indicating less distinction between the transcriptomes of these phases for BSFs. As observed in PCFs, early G1 and unlabelled BSF transcriptomes overlapped significantly (*Figure 3B*). DE analysis between these two phases identified 16 genes (adjusted p-value <0.05), yet none reaching a FC cut-off of >1.5 (*Supplementary file 3*). Of these, three were also DE between early G1 and unlabelled PCFs: a putative ribosomal protein S9/S16 (Tb427_070014300), a hypothetical protein (Tb427_010013900), and an RPA subunit (Tb427_050022800). The proportion of cells in each phase was similar between fresh and frozen samples (*Figure 3C*), as well as phases defined by DNA content (*Figure 3D*).

Using Cyclum to infer pseudotime during the cell cycle (*Figure 3E*), also indicated S and G2M BSF cells were less distinct in their transcriptome than in PCFs at this cell cycle transition. DE analysis over pseudotime identified 1864 CCR transcripts (FDR adjusted p-value <0.01, FC >1.5) with dynamic expression during the cell cycle (*Figure 3F*, *Supplementary file 5*). A remarkably similar proportion of genes peaking in each phase was found for BSF compared to PCF. In BSFs, 76 (4.08%) genes had highest expression in early G1, 588 (31.55%) in late G1, 678 (36.37%) in S phase, and 522 (28.00%) in G2M (*Supplementary file 5*, *Figure 3F*).

Proteomics data across the cell cycle are not currently available for BSFs; yet, 13.83% (122 of 882 detected) and 12.54% (155 of 1236 detected) of the BSF CCR transcripts were identified as CCR in PCF proteomics datasets from Benz (*Figure 2—figure supplement 4*) and Crozier (*Figure 3G*), respectively. The expression pattern of these common CCR genes, largely following the same pattern as PCF, with transcripts peaking prior to protein levels.

To investigate CCR proteins directly in BSFs, the top most significant genes with transcripts peaking in late G1, S, and G2M phase were tagged at the N- or C-terminus with the fluorescent epitope tag mNeonGreen (mNG) using CRISPR/Cas9 (*Beneke et al., 2017*). The top 2 CCR genes to peak in late

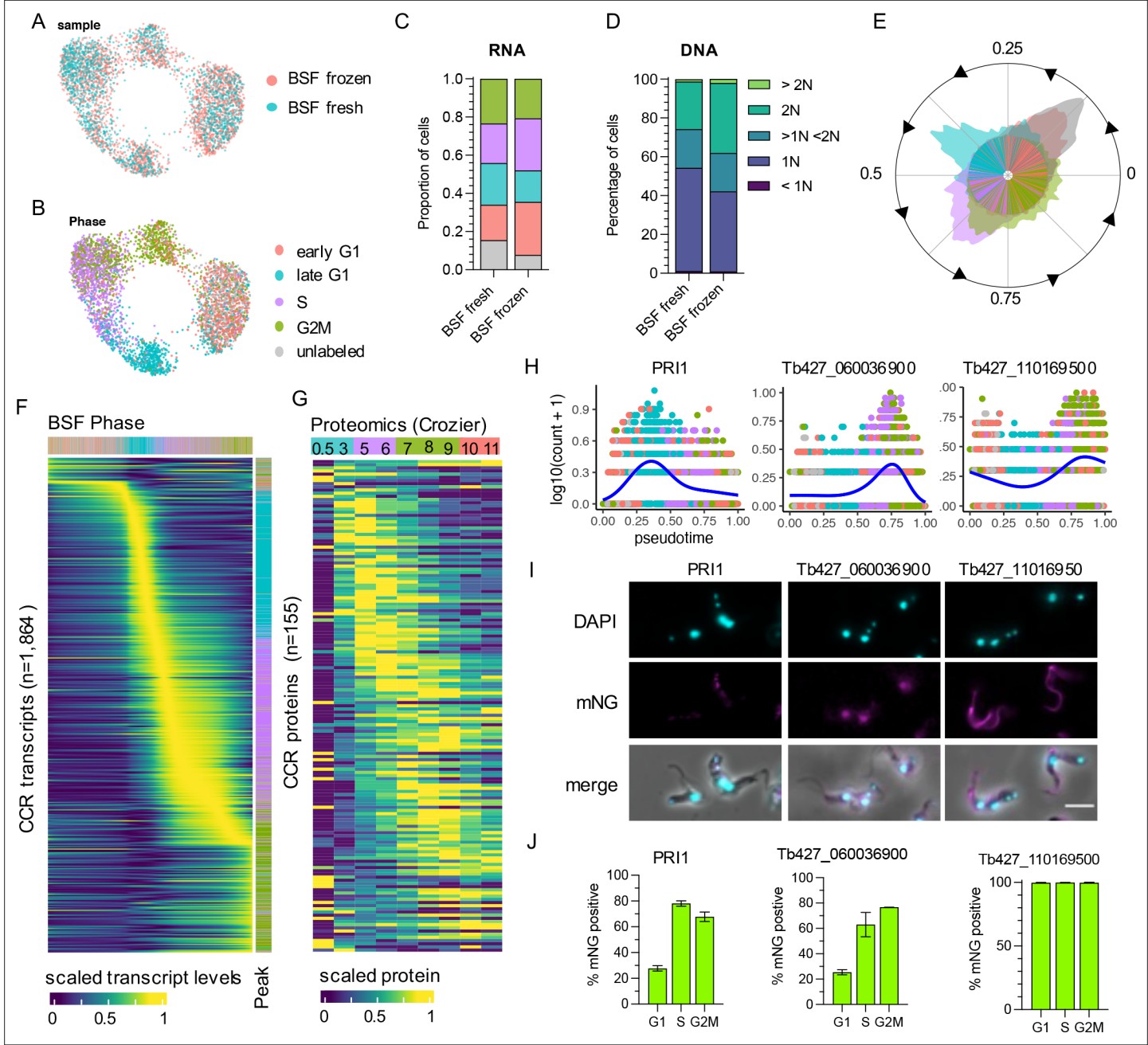

**Figure 3.** The cell cycle transcriptome of bloodstream form (BSF) *T. brucei*. (**A**) Unifold manifold approximation and projection (UMAP) plot of integrated BSF transcriptomes from fresh (blue) and frozen (red) samples. (**B**) UMAP of BSF transcriptomes coloured by inferred cell cycle phase. (**C**) Proportion of cells assigned to each phase by transcriptomics as in B. Legend as in B. (**D**) Proportion of cells with DNA content assessed by flow cytometry. (**E**) Histogram of transcriptomes arranged in pseudotime (anti-clockwise) representing cell cycle progression. Each line in inner circle indicated one transcriptome coloured by phase as in B. Outer circle histogram of showing number of cells at each point in pseudotime (0–1). (**F**) Scaled transcript levels of cell cycle regulated (CCR) genes (rows), ordered by peak time, plotted across transcriptomes (columns) ordered in pseudotime. Top annotation indicates cell phase, right annotation indicates phase with highest expression of each gene. (**G**) Scaled protein abundance for 137 genes identified as CCR by Crozier et al., plotted in the same order as F. Time points are indicated in top annotation, coloured by the most enriched cell cycle phase for each sample. (**H**) Transcript counts of three of the top CCR genes (*y*-axis) plotted across pseudotime (*x*-axis). Each dot shows one transcriptome coloured by phase as in B. Blue line shows smoothed expression level across pseudotime. (**I**) Fluorescent microscopy imaging of mNeonGreen (mNG) tagged top CCR proteins. DAPI (4′,6-diamidino-2-phenylindole) staining of DNA (cyan) and mNG fluorescence (magenta) are shown for the three genes as well as merged with DIC (merge). Scale bar = 10 μm. (**J**) The percentage of cells positive for mNG as detected by flow cytometry analysis. For each gene, counts are separated by cell cycle phase, inferred by DNA content detection (G1 = 2C, S = >2C < 4C, G2M = 4C). Error bars indicate the standard deviation from the mean of three (Tb427_080028700 and Tb427_110169500) or two (Tb427_060036900) biological replicates.

*Figure 3 continued on next page*

*Figure 3 continued*

The online version of this article includes the following source data and figure supplement(s) for figure 3:

**Figure supplement 1.** Analysis of mNG tagged protein expression in bloodstream forms (BSFs).

**Figure supplement 1—source data 1.** Original gel images.

G1 were MORN repeat-containing protein 1 (MORN1, Tb427_060051900) and mitochondrial DNA primase, Pri1 (Tb427_080028700) (*Figure 3H*). MORN1 has been localised by immunofluorescence previously, revealing the protein is part of the specialised trypanosome bilobe, cytoskeletal structure located close to the flagella pocket (*Esson et al., 2012*; *Morriswood and Schmidt, 2015*). PRI1 was found previously to locate to the antipodal sites flanking the mitochondrial kDNA in PCFs (*Hines and Ray, 2010*). Using fluorescence microscopy, we found that tagged mNG::PRI1 (N-terminal tag), akin to the observations in PCFs, also localises to the flanking sides of kDNA in BSFs (*Figure 3I*). Flow cytometry analysis was used to compare expression of mNG::PRI1 to cell cycle phase, inferred using DNA content (*Figure 3J* and *Figure 3—figure supplement 1*). While 27.71% G1 cells were detected as expressing mNG::PRI1, this increased to 78.06% in S phase cells and 67.73% of G2M phase cells. Additionally, fluorescence intensity peaked in S phase cells (*Figure 3—figure supplement 1*). Thus, a delay between protein and transcript levels are also notable for PRI1, with transcripts peaking in late G1 (*Figure 3H*), but protein in S phase (*Figure 3J* and *Figure 3—figure supplement 1*).

The top S phase peaking transcripts were two genes encoding histone H2B (Tb427_100112400 and Tb427_100112200), followed by Tb427_060036900 which encodes a hypothetical protein of no known function (*Figure 3H*). N-terminal tagging of this gene was unsuccessful, but C-terminal tagging resulted in viable clones. Fluorescence microscopy revealed nuclear localisation, including in post-mitotic cells where both nuclei contained fluorescent protein (*Figure 3H*, *Figure 3—figure supplement 1*). Flow cytometry revealed 25.44% of G1 phase BSFs expressed Tb427_060036900::mNG, increasing to 63.04% and 76.72% of S and G2M phase cells, respectively (*Figure 3J*), in keeping with cyclic transcript levels expression increasing in S and G2M phase (*Figure 3H*).

The most significant gene peaking in G2M was Tb427_110169500, which also encodes a hypothetical protein (*Figure 3H*). N-terminal tagging revealed this protein is expressed in all cycle phases (*Figure 3I, J*) and localises to both the old and newly developing flagellum (*Figure 3I*, *Figure 3—figure supplement 1*). Transcript levels increase as cells progress from into S and peak in G2M, perhaps to meet increased protein requirement as the new flagella develops. Slightly increased fluorescence of mNG was evident for G2M cells compared to G1 (*Figure 3—figure supplement 1*).

## Common CCR transcripts in BSF and PCF forms

BSF and PCF CCR transcripts were compared to identify 1013 genes classed as CCR in both forms using a threshold of adjusted p-value <0.01 and FC >1.5 (*Figure 4—figure supplement 1*, *Supplementary file 6*). Expression patterns appeared to show greater coordination in the early stages of the cell cycle, whereas patterns in the S and G2M phase showed greater variability between forms (*Figure 4A, B*). 83.12% (842) of the common CCR genes showed highest transcript levels in the same cell cycle phase, and 16.19% (164) peaked in neighbouring phases (*Figure 4B*).

Genes were classified based on the phase with highest expression in the BSF cell cycle (*Supplementary file 4*), and Gene Ontology (GO) analysis was performed to find biological processes associated with each set of genes (*Figure 4D*, *Supplementary file 6*). Few GO terms were enriched for early G1 genes, as only nine genes peak in this phase, and all of these are either labelled as 'hypothetical' or have been assigned descriptions based on putative functional domains. All GO terms, including 'rRNA processing', relate to one gene, Tb427_110120100, which shares sequence homology with UTP21, a component of the small-subunit processome (*Barandun et al., 2017*).

In late G1, several genes associated with 'protein phosphorylation' are evident, including known cell cycle-associated genes such as CRK2 (*Mottram and Smith, 1995*; *Tu and Wang, 2005*; *Tu and Wang, 2004*), aurora kinase 3 (AUK3; *Jones et al., 2014*; *Tu et al., 2006*), and Wee1-like kinase (*Boynak et al., 2013*). Fourteen late G1 genes related to the term 'DNA replication': 5 genes encoding components of the MCM DNA replication licensing complex (MCM2, 4–7), an MCM10 homolog, 5 DNA polymerases, 2 DNA primase subunits and the DNA synthesis factor RNR1 (ribonucleoside-diphosphate reductase large chain). DNA repair protein RAD9 and recombination helicase proteins

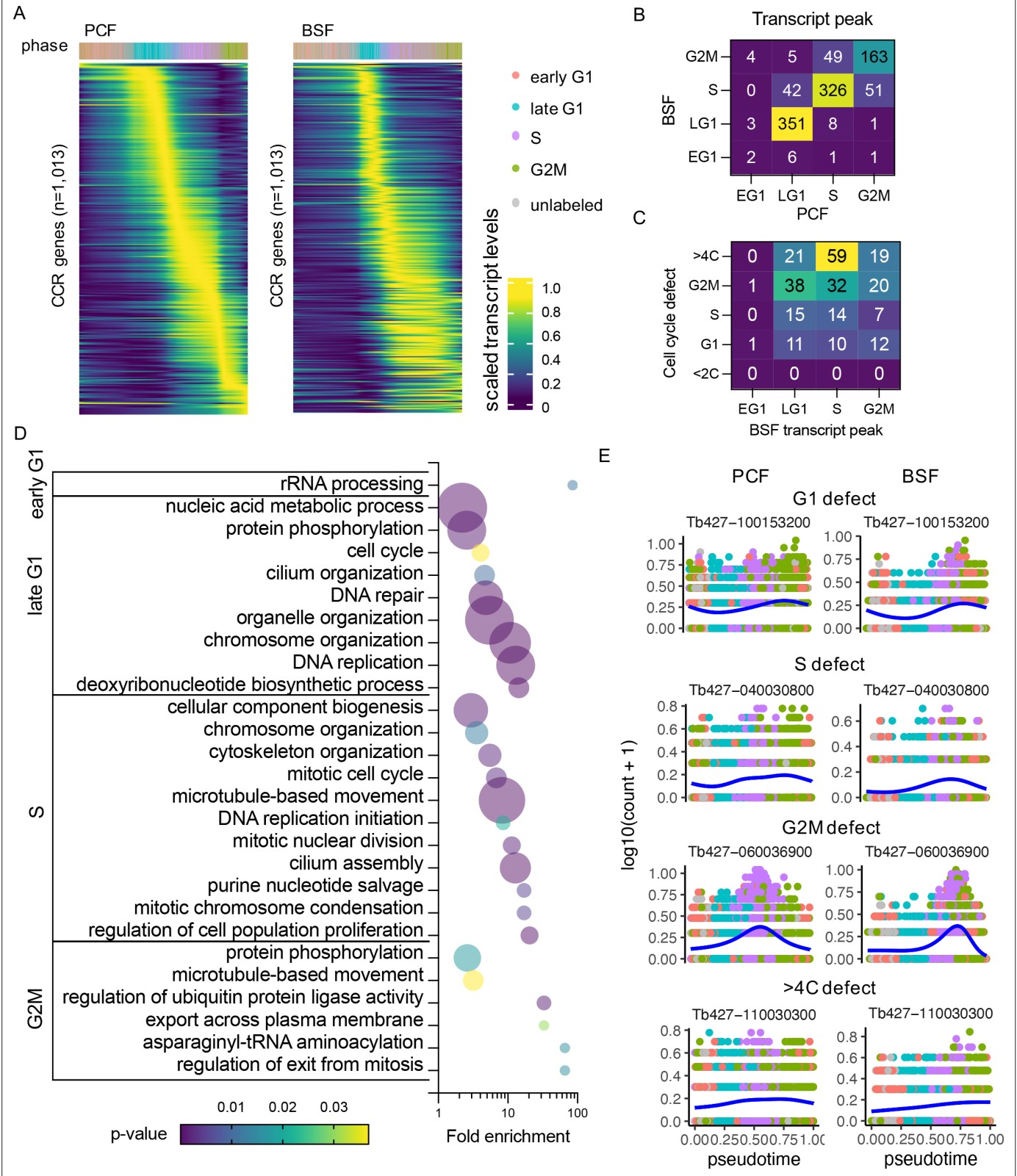

**Figure 4.** Common cell cycle regulated (CCR) transcripts of bloodstream form (BSF) and procyclic form (PCF) *T. brucei*. (**A**) Scaled transcript levels of common CCR genes (rows), ordered by peak time and plotted across transcriptomes (columns) ordered in pseudotime of PCF (left) and BSF (right). Genes are ordered by peak time in the PCF cell cycle in both cases for direct comparison. Top annotation indicates cell cycle phase. (**B**) Number of genes peaking in each cell cycle phase for PCF (*x*-axis) and BSF (*y*-axis) transcriptomes. (**C**) Number of genes peaking in each BSF phase (*x*-axis) linked

*Figure 4 continued on next page*

*Figure 4 continued*

to a cell cycle defect (*y*-axis) in RIT-seq screen of BSFs by *Marques et al., 2022*. (**D**) Gene Ontology (GO) terms associated with common CCR grouped by peak phase in the BSF cell cycle. Fold change of detected genes is plotted on *x*-axis, points are sized by the number of genes and coloured by p-value. (**E**) Transcript levels of the most significantly differential expression (DE) gene associated with each cell cycle defect category (G1, S, G2M, and >4C). Counts per cell (*y*-axis) are plotted across PCF (left) and BSF (right) pseudotime (*x*-axis), coloured by phase as in A. Blue line shows smoothed expression level across pseudotime.

The online version of this article includes the following figure supplement(s) for figure 4:

**Figure supplement 1.** Cell cycle regulated (CCR) genes in procyclic form (PCF) and bloodstream form (BSF) classified by adjusted p-value and FC.

**Figure supplement 2.** Smoothed gene expression pattern of cell cycle regulated (CCR) cyclins.

PIF1, 2, and 5 show similar regulation, are associated with 'DNA repair' and, the in case of the helicase proteins, 'telomere maintenance' GO terms.

S phase is associated with GO terms 'DNA replication initiation', due to peak expression of CDC45 (cell division cycle 45) and another MCM component, MCM3. Together with the GINS complex, MCM2–7 and CDC45 form the replicative helicase CMG complex (*Dang and Li, 2011*) that is activated only in S phase. Two genes encoding putative components of the condensin complex (CND1 and CND3) are predicted to have roles in 'mitotic chromatin condensation' and 'mitotic nuclear division'. Mitotic cyclin CYC6 is clearly CCR in both forms and peaks at the S–G2M transition (*Figure 4—figure supplement 2*), while a second mitotic cyclin, CYC8, peaks during late G1 in both life cycle stages (*Figure 4—figure supplement 2*). Kinetoplastid membrane protein 11 (KMP11-2), a known positive regulator of cytokinesis in both BSF and PCF (*Li and Wang, 2008*) together with two paralogs, KMP11-1 and KMP11-5, also peaked in S phase.

In G2M phase, AUK1 and AUK2 transcripts are at their highest levels which, along with the kinetochore phosphorylating kinase KKT10 (*Ishii and Akiyoshi, 2020*) (kinetoplastid kinetochore protein 10) and five other kinases, are enriched for the term 'protein phosphorylation'. Six genes associated with 'microtubule based process' were upregulated in G2M, including two putative kinesins (one of which, KIN-F, is known to localise to the spindle during mitosis; *Zhou et al., 2018*) and KLIF (kinesin localising to the ingress furrow), which is required for cleavage furrow ingression during cytokinesis (*Zhou et al., 2022*). CDC20 transcripts are highest in G2M and is associated with the 'regulation of ubiquitin protein ligase activity' term; yet, there is no evidence of CDC20 acting on the Anaphase Promoting Complex/Cyclosome (APC/C), at least in PCFs (*Bessat et al., 2013*). Other G2M-associated genes include a putative homolog of *S. cerevisiae* CDC14, which has several roles in regulating mitosis (*Manzano-López and Monje-Casas, 2020*).

RNA-binding proteins (RBPs) drive UTR-mediated mRNA level modulation in *T. brucei* (*Clayton, 2019*). We find 18 CCR genes (*Supplementary file 6*), in both forms that have documented RNA-binding domains (RNA-recognition motif, CCCH class zinc finger, *Kramer et al., 2010* and Pumilio domain) or experimentally identified in BSFs (*Lueong et al., 2016*). These include pumilio domain protein PUF9 that peaks in S phase in both forms, as previously noted in PCFs (*Archer et al., 2009*). In agreement with *Archer et al., 2009*, we find the four target mRNAs increase during S phase in both forms (*Supplementary file 6*). Other common RBPs do not have documented target mRNA, but vary in where they peak in the cell cycle and may act as key regulators in the cell cycle.

Recently, a genome-scale phenotypic genetic screen (RNA Interference Target sequencing, RIT-seq) was performed to identify genes associated with a cell cycle defect when transcripts were depleted by RNAi in BSF *T. brucei* (*Marques et al., 2022*). After induction of RNAi, cells from each cell cycle phase (G1, S, and G2M) were isolated based on their DNA content using FACS; sub-diploid (<2C) and over-replication (>4C) populations were also isolated and analysed. Analysis of each pool revealed depletion of which genes had led to an enrichment of parasites in each population, associating a cell cycle defect to 1198 genes (16.63% of those investigated). Of the 1013 common CCR genes identified here by scRNA-seq analysis, 260 (25.67%) were shown to have a cell cycle defect using the same threshold as Marques et al. (*Figure 4C*, *Supplementary file 6*). The peak transcript expression phase of these genes showed low association with phenotype defect, although 22.69% of genes with highest expression in S phase resulted in a >4C defect, indicating these S phase genes are required for correct genome replication and its control.

As mechanisms of cell cycle regulation and cyclical transcript changes are largely conserved across the eukaryotes, we hypothesised that genes with CCR transcript levels in *T. brucei* are more likely to be

conserved. To investigate, we extracted the 819 orthogroups which contained the common CCR regulated genes and compared the orthogroup conservation across 44 kinetoplastid proteomes, including trypanosome and leishmania species (*Oldrieve et al., 2022*). CCR orthogroups were conserved across significantly (p < 0.0001) more proteomes (mean of 41.66 out of 44), compared all orthogroups of the *T. brucei* Lister427 proteome (*Müller et al., 2018*) (mean of 18.44), and a random subset of 1000 orthogroups (mean 18.43) (*Figure 4—figure supplement 1*). More proteins per orthogroup were also present across the kinetoplastid species for orthogroups containing CCR genes (mean of 50.79 proteins per orthogroup) compared to all orthogroups (mean of 23.31 proteins per orthogroup) and the random subset (mean of 22.45 proteins per orthogroup) (*Figure 4—figure supplement 1*). Of the highly conserved common CCR genes, 365 genes are described as encoding 'hypothetical' proteins (9.69% of the total hypothetical protein encoding genes located in the core chromosomes), indicating they may have central unknown roles in the kinetoplastida cell division cycle. Of these, 61 had a cell cycle defect identified by Marques et al.; depletion of 9 led to increased BSF in S phase, 23 in G2/M, 14 in G1, and 15 in >4C. The transcript levels for most significant genes for each defect are plotted across the PCF and BSF cell cycle (*Figure 4E*).

## Unique CCR transcripts in PCF and BSF

Of the CCR transcripts in PCF, 540 were only significant in this form and showed varied expression across all phases of the cell cycle (*Figure 5A*, *Supplementary file 7*). GO term enrichment (*Figure 5B*, *Supplementary file 7*) of these genes uncovered terms including 'lipid metabolic process' attributed to 9 genes encoding putative proteins, including one encoding a putative triacylglycerol lipase which peaks in S phase, and a putative C-14 sterol reductase, for which transcripts are highest in late G1 (*Figure 5—figure supplement 1*). Genes relating to 'ribonucleoprotein complex biogenesis' include ribosome production factor 2 (RPF2), which is part of the 5S ribonucleoprotein (RNP) complex in PCFs (*Jaremko et al., 2019*), and 20S-pre-rRNA D-site endonuclease, NOB1, which matures the 3′ end of 18S rRNA (*Kala et al., 2017*; *Figure 5—figure supplement 1*). "DNA replication" associated genes include replication factors RPA2 and putative, RPC3 (*Figure 5—figure supplement 1*), both of which show growth defects in PCFs (*Jones et al., 2014*; *Rocha-Granados et al., 2018*).

The 851 uniquely DE genes in BSFs also showed varied expression dynamics over the cell cycle (*Figure 5C*). GO term analysis highlighted 52 genes linked to the term 'phosphorus metabolic process', 12 to 'carbohydrate metabolic process' and 7 specifically to 'glycosyl compound metabolic process'. These metabolic associated genes include 24 components of the glycolysis/gluconeogenesis pathway, including glucose-6-phosphate isomerase (PGI), phosphoglycerate kinase (PGKC), and triosephosphate isomerase (TIM) (*Figure 5—figure supplement 2*). Enzymes linked to the term 'phosphorylation' included Repressor of Differentiation Kinases 1 (RDK1) and 2 (RDK2), both of which repress differentiation from BSF to PCFs (*Jones et al., 2014*), and a pseudokinase linked to slowed growth in BSFs when depleted in a kinase-specific RIT-seq screen (*Jones et al., 2014*). Interestingly, RDK1 and RDK2 show inverse expression patterns, with RDK1 transcripts at lower levels in late G1 before rising as the cell cycle progresses through S, G2M, and back into early G1 (*Figure 5—figure supplement 2*). Four genes are linked to 'DNA recombination': RPA1, KU80, RAD51, and RecQ helicase, each with a varied expression pattern (*Figure 5—figure supplement 2*). RPA1 transcripts peaked in late G1, in keeping with the finding at the CfRPA1 mRNA peaks at the G1–S boundary in *Crithidia fasciculata* (*Pasion et al., 1994*). Two histone-lysine *n*-methyltransferases, DOT1A and DOT1B, are also significantly CCR only in BSFs with these thresholds. However, both DOT1s do have a similar smoothed expression pattern in BSF and PCF (*Figure 5—figure supplement 2*) despite neither gene reaching the required thresholds in this analysis to be considered CCR in PCF (*Supplementary file 7*).

In addition, cyclin encoding gene CYC4 was CCR only in BSFs where transcripts peak in G2/M (*Figure 4—figure supplement 2*). A previously un-investigated cyclin domain-containing gene was uncovered in this analysis, Tb427_110012500, which peaks between early and late G1 phases of BSFs (*Figure 4—figure supplement 2*). In contrast, CYC9 is significant only in PCFs, yet shows only a slight increase in transcript levels as the cell cycle progresses to G2/M (*Figure 4—figure supplement 2*).

In addition to common RBPs, we find 12 uniquely regulated in the PCF cell cycle and 37 in BSFs only (*Supplementary file 6*). Master regulator of slender to stumpy differentiation, ZC3H20 (*Liu et al., 2020*; *Cayla et al., 2020*), displayed cycling transcript levels peaking in G2/M uniquely in the BSF, along with ZFP1 which is involved in specific repositioning of the kDNA genome during this

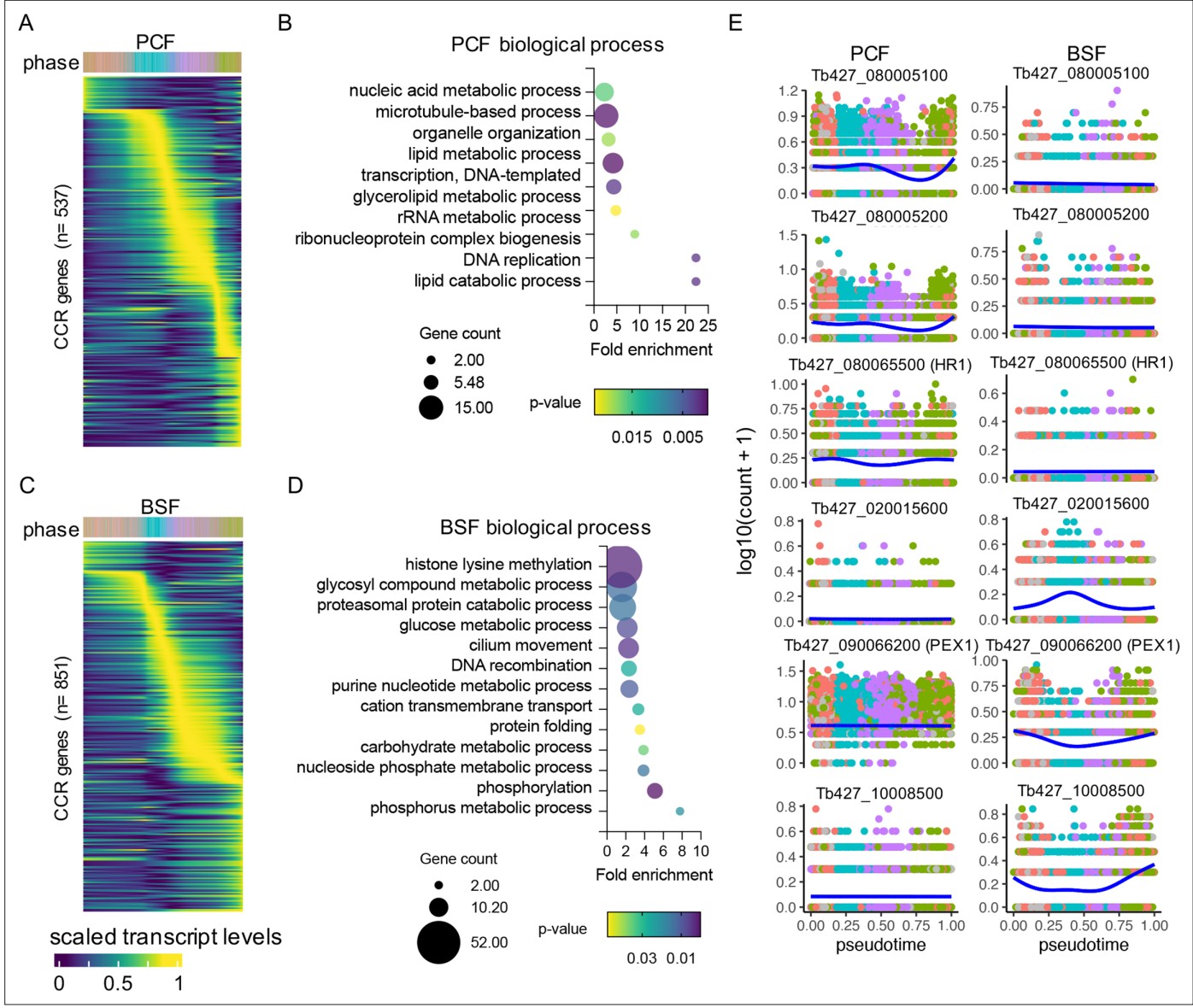

**Figure 5.** Unique cell cycle regulated (CCR) transcripts of bloodstream form (BSF) and procyclic form (PCF) *T. brucei*. (**A**) Scaled transcript levels of unique CCR genes (rows), ordered by peak time, plotted across transcriptomes (columns) ordered in pseudotime across PCF cell cycle. Top annotation indicates cell phase. (**B**) Gene Ontology (GO) terms associated with CCR genes unique to PCFs. Fold change of detected genes is plotted on *x*-axis. Points are sizes by number of genes and coloured by p-value. (**C**) Scaled transcript levels for CCR unique to BSF cells cycle, as in **A**. (**D**) GO terms associated with unique BSF CCR genes, as in **B**. (**E**) Transcript levels of six genes with strong association bias to one life cycle form cell cycle. Counts per cell (*y*-axis) are plotted across PCF (left) and BSF (right) pseudotime (*x*-axis), coloured by phase as in **A**. Blue line shows smoothed expression level across pseudotime.

The online version of this article includes the following figure supplement(s) for figure 5:

**Figure supplement 1.** Smoothed gene expression pattern of genes associated only with the procyclic form (PCF) cell cycle.

**Figure supplement 2.** Smoothed gene expression pattern of genes associated only with the bloodstream form (BSF) cell cycle.

developmental transition (*Hendriks and Matthews, 2005*) and also peaks in transcript levels in G2/M phase. ZC3H21 in contrast is only expressed in the PCFs (*Liu et al., 2020*) and here peaks in expression during the G2/M phase. *Liu et al., 2020* identified 28 target mRNAs bound by both ZC3H20 and ZC3H21 in PCFs, of which we find three are also CCR in the PCF with two peaking in G2/M (Tb427_100143800 encoding a hypothetical protein, and Tb427_110041100 encoding a predicted thiamine pyrophosphokinase) and one in S phase (Tb427_080083400 encoding a transmembrane domain-containing protein). DRBD13 was only CCR in PCFs (S phase peak) and is essential for viability

and the expression of PCF surface proteins. Notably, the metacyclogenesis factor (*Kolev et al., 2012*) and target of DRBD13 (*Jha et al., 2015*), RBP6, is also CCR, but only in BSFs where the transcripts peak in S phase. RBP10 is another life cycle regulator only expressed as a protein in the BSF (*Wurst et al., 2012*). Although, we do not find the transcripts encoding RBP10 to be CCR, of the 260 RBP10-target mRNAs identified by *Mugo and Clayton, 2017*, 61 were CCR in the BSF cell cycle (*Supplementary file 5*), with an enrichment of S phase peaking transcripts (31/61, 50.8%) compared to all CCR transcripts (678/1864, 36.4%). The remaining CCR RBP10-targeted were spread across the early G1 (4/61, 6.55%), late G1 (14/61, 22.95%), and G2/M phases (12/61, 19.67%). Therefore, although there is an enrichment for S phase-associated transcripts, RBP10 appears to regulate mRNA with varying expression patterns as well as repressing PCF-associated genes (*Mugo and Clayton, 2017*).

Thresholds for classifying CCR were selected based on the detection of previously identified cell cycle phase markers (*Archer et al., 2011*) in this scRNA-seq analysis in PCFs (*Figure 2—figure supplement 2*). Genes were only considered common to each form if both adjusted p-value <0.01 and FC >1.5 thresholds were satisfied in both forms, otherwise they are considered unique to one form. The transcript dynamics for six of the genes showing greatest difference in p-value between forms (*Figure 4—figure supplement 1*) are plotted *Figure 5e*. These include 5 genes encoding hypothetical proteins and one encoding putative peroxisomal biogenesis factor 11 (PEX11), which is only CCR in BSF and peaks in G2M and early G1 phases. If only p-values are considered when comparing BSF and PCF CCRs, 1513 genes are considered common to both forms, 522 unique to PCFs and 366 to BSFs. If using the FC threshold only to compare genes, 1804 are considered CCR in both, 271 in PCFs and 325 in BSFs. All comparisons are available in *Supplementary file 7*.

## Discussion

In this work, we provide the CCR transcriptomes of both BSF and PCF *T. brucei*, generated from asynchronously replicating populations. Computational reconstruction of the cell cycle with individual transcriptomes allowed us to ascertain the extent to which each gene's transcript levels follow the periodic waves of the cell cycle and map their dynamic patterns. Comparison between transcript expression patterns and previously published protein abundance changes identified a relative delay in peak levels for transcript and protein for at least 50% of the genes that could be compared. Comparing BSF and PCF cyclic transcriptomes identified a common set of highly conserved CCR genes, enriched for known cell cycle-related genes and, thus, likely novel regulators of cell cycle in kinetoplastidae. Intriguingly, a key difference between forms appears at the S–G2 transition where the gene expression switch associated with these phases is more tightly regulated in PCFs compared to BSFs.

### Cryopreservation as a method to capture transcriptomes

In addition to specific analysis of the cell cycle, we provide evidence that scRNA-seq analysis of cryopreserved parasites is feasible without detrimentally altering the transcriptome of parasites providing a methodological development likely to be of utility in multiple scRNA-seq studies.

scRNA-seq has proved a powerful method for investigating trypanosome parasites, yet its implementation is still restricted by high cost and the need to isolate live parasites (*Briggs et al., 2021b*). We previously attempted to perform Chromium scRNA-seq with BSF *T. brucei* fixed in methanol, but this resulted in low transcripts detection per cell preventing meaningful analysis (*Briggs et al., 2021b*). *Howick et al., 2022* employed plate-based method SMART-seq2 to analyse *T. brucei* isolated from tsetse flies (*Howick et al., 2022*). This method generally generates higher coverage transcriptomes with full-length transcripts, but at lower through-put than droplet-based methods as this technique is limited by the use of multi-well plates to isolate cells. These authors compared the transcriptomes derived from live *T. brucei* to those prepared with two preservation methods: dithio-bis(succinimidyl propionate) (DSP) fixation or Hypothermosol-FRS preservation. Although both methods resulted in the recovery of transcriptomes comparable with live cells, conclusions could not be drawn about the impact of these methods as each was applied to samples from different experimental time points, confounding comparisons (*Howick et al., 2022*).

Here, we compare Chromium generated transcriptomes from PCF and BSF prepared immediately from in vitro culture to those carefully cryopreserved with 10% glycerol and then thawed slowly. Comparison between fresh and frozen samples revealed few significant changes to gene expression,

both when considering expression averaged across the population and between individual transcriptomes. Only one gene showed altered transcript levels between conditions for both forms: fructose-bisphosphate aldolase (ALD; Tb427_100060400), a component of the glycolytic pathway, was downregulated after freezing. ALD protein is detectable in both BSF (*Barbosa Leite et al., 2020*) and PCF (*Jones et al., 2006*; *Vertommen et al., 2008*), but has higher transcript levels in BSFs in other studies (*Siegel et al., 2010*; *Kabani et al., 2009*; *Queiroz et al., 2009*; *Jensen et al., 2014*). It is unclear here whether the temperature changes, or use of glycerol (which BSFs have been demonstrated to use as a substrate in gluconeogenesis; *Kovářová et al., 2018*) in the freezing/thawing procedure triggered decreased ALD transcripts. In PCFs, PAG1–5 were all upregulated in cryopreserved transcriptomes. PAGs are not essential for differentiation from BSF to PCF, but mRNA levels of PAGs 1–3 were transiently upregulated during the BSF to PCF differentiation, trigged by reducing temperature and addition of cis-aconitate (*Haenni et al., 2006*). PAG4 and PAG5 were not analysed in that study as levels were not detectable by blotting (*Haenni et al., 2006*). Hence, PAG transcript level changes are likely to be induced by the temperature change during cryopreservation.

Other than these isolated changes, we could not find significant differences between fresh and frozen transcriptomes in either form. Furthermore, freezing had little effect on the transcript recovery per cell, and samples could be fully integrated to study the biological process of interest without confounding results. Thus, cryopreservation is an appropriate method of storing *T. brucei*, and likely related parasite species, prior to scRNA-seq.

## Global cell cycle analyses

Previously, profiling of the PCF cell cycle transcriptome relied on centrifugal elutriation or serum starvation (*Archer et al., 2011*). In both cases, parasites were returned to normal culture conditions and RNA was extracted from discrete time points for sequencing. Although time points were clearly enriched for cell cycle phases, samples still contained mixed populations to varying degrees. Now technological and analytical advances have made it possible to avoid these potentially stress inducing methods by performing scRNA-seq directly on asynchronous mixed populations, with their cell cycle phases then resolved computationally. We applied pseudotime inference and DE methods to profile cyclical transcript changes, rather than directly comparing discretely grouped phases. Although it is likely that genes with low transcript levels are missed in this analysis, as sensitivity of scRNA-seq is lower than bulk-RNA-seq (*Lähnemann et al., 2020*; *Qiu, 2020*; *Mou et al., 2019*), 1550 genes with dynamic transcript level changes reflective of the cell cycle were identified, including 1151 which had not been identified by bulk analysis. These CCR genes include new transcriptional markers of each phase, including those clearly distinguishing late G1 phase PCFs from early G1 phase parasites, for which previously identified early G1 markers were insufficient for labelling (*Figure 2B, F*).

scRNA-seq also allowed the characterisation of the BSF cycling transcriptome for the first time. Using the same significance thresholds, we identified 1864 genes with CCR transcript levels, 1013 of which were also identified as CCR in the PCF cell cycle. The additional CCR genes identified only in the BSF included those linked to glycolysis, which BSFs rely on to generate ATP from the glucose energy source in the mammal (*Coley et al., 2011*). Interesting, the knockdown of 11 glycolysis-associated genes was linked to cell cycle arrest in G1 (*Marques et al., 2022*) and so further investigation may unveil if BSFs use glycolysis activity levels as a signal for re-entering the cell cycle during G1. Other genes uniquely CCR in BSFs include DNA recombination factors RecQ helicase and Rad51. RECQ functions to repair DNA breaks, including at the subtelomeric sites of variant surface glycoprotein (VSG) expression (*Devlin et al., 2016*), and is hypothesised to limit strand exchange during homologous recombination (HR) reactions at this site (*Faria et al., 2022*). HR at *VSG* expression sites is central to antigenic variation required for evasion of the mammalian immune system and so survival of BSF parasites. RAD51 is central to the recombination of previously silent *VSGs* into the transcribed *VSG* expression site to allow expression of a new VSG on the parasite surface (*McCulloch and Barry, 1999*). It is hence possible that these genes show higher CCR expression in BSFs due to their role in antigenic variation-associated HR events, which may be triggered by DNA replication-associated damage (*Devlin et al., 2017*) and so require specific expression timing in the cell cycle.

A further notable difference between forms is the clear distinction of S and G2M phases in PCFs, compared to much less apparent separation in BSFs when using both UMAP and independent pseudotime inference approaches. This indicates that the switch in gene expression associated with the S–G2

transition is much more discrete or tightly regulated in PCFs than BSFs. Comparing the expression patterns of shared CCR genes in each form (*Figure 4*) further highlights that expression patterns of the G1 and S phase genes are highly comparable between forms, whereas after S phase the timing of gene expression is far less synchronised. Human cells display tight regulation of the S–G2 transition, with the mitotic gene network only expressing after the end of S phase (*Saldivar et al., 2018*). Here, ATR kinase remains in its active form throughout S phase and cells only progress to G2, and upregulate the associated gene programme, upon ATR inactivation at the end of S to ensure complete genome replication prior to mitosis (*Saldivar et al., 2018*). In the absence of ATR, human cells activate DNA replication origin firing aberrantly, and undergo premature and defective mitosis (*Eykelenboom et al., 2013*). Interestingly, ATR activity in *T. brucei* is required for normal S phase progression in both PCFs (*Marin et al., 2020*) and BSFs (*Black et al., 2020*), yet the proteins' role in the S–G2 transition differs dramatically between forms. In BSFs, ATR depletion is lethal and within 24 hr increases the proportions of S and G2M phase parasites, as well as aberrant cells resulting from premature mitosis and cytokinesis events, indicating a putatively similar role to human ATR (*Black et al., 2020*). Yet in PCFs, ATR knockdown has little effect on the cell cycle indicating PCFs mostly undergo the S–G2 transition and complete mitosis and cytokinesis correctly without ATR activity (*Marin et al., 2020*). Thus, as highlighted by scRNA-seq investigations here, PCFs and BSFs appear to, at least partially, regulate the S–G2 transition differently. Why BSFs would not require the same level, or mechanism, of regulation of this transition is currently unclear. Intriguingly however, even in the presence of persistent DNA damage BSFs will continue to replicate DNA and proliferate (*Glover et al., 2019*). As BSFs require DNA damage at *VSG* expression sites to trigger HR and VSG switching, it is plausible that BSF allows continuation to G2 in the presence of DNA damage acquired during S phase, which could then be repaired to facilitate *VSG* recombination event in subsequent phases. Indeed, Rad51 transcripts peak at the S–G2M transition, a pattern not observed in PCFs (*Figure 5—figure supplement 2c*), and in BSFs Rad51 foci form mainly in G2/M phase parasites (*Glover et al., 2008*).

## Conservation of cell cycles between life stages

Interrogating the shared and unique CCR transcriptomes is likely to unveil new insights into *T. brucei* cell cycle regulation, for example by assess expression patterns of cyclins. In *T. brucei*, 13 cyclins have been investigated and several cyclin–CRK-binding pairs have been documented (*Hammarton, 2007*; *Lee and Li, 2021*; *Li, 2012*; *Wheeler et al., 2019*). Notably, we find just two cyclins with strong CCR transcript dynamics in both forms, CYC8 and CYC6. CYC6 binds CRK3 (*Hammarton et al., 2003*) and is well characterised as essential for mitosis (*Hammarton et al., 2003*; *Li and Wang, 2003*; *Hayashi and Akiyoshi, 2018*) in both forms, correlating with expression levels detected here at the S–G2M transition. CYC8 instead clearly peaks during late G1, despite RNAi depletion leading to a slight increase in G2/M cells in PCFs (*Li and Wang, 2003*). Thus, although both cyclins have roles in G2/M, CYC8 peak earlier in the cell cycle and is followed by the gradual rise in CYC6. Despite these different patterns, transcripts of both cyclins are reported to be bound by the RNA-binding protein RBP10 (*Mugo and Clayton, 2017*), highlighting that steady-state RNA levels are likely regulated by multiple factors beyond the individual RBPs. Additionally, RBP10 is not expressed in PCF (*Wurst et al., 2012*; *Dejung et al., 2016*), and so how matching cyclic expression patterns are regulated in both forms is unclear. CYC8 transcripts were previously identified as enriched in G1 (*Archer et al., 2011*), yet protein levels were undetectable (*Crozier et al., 2018*). Protein levels of CYC6 have been documented as CCR (*Crozier et al., 2018*), yet previously, CYC6 transcripts were not recorded as CCR (*Archer et al., 2011*), exemplifying the power of scRNA-seq over bulk transcriptomics. ScRNA-seq analysis finds only a slight CYC9 transcript increase in G2M, and only in PCFs. Yet, in BSFs CYC9 transcript depletion results in a clear cytokinesis defeat (*Monnerat et al., 2013*). Thus, transcript FC does not necessarily correlate with functional significance, as was also noted when comparing CCR genes to cell cycle defects profiled by the genome-scale screen in BSFs (*Marques et al., 2022*). Results of CYC9 RNAi depletion in PCFs are currently conflicting, possibly due to differences in knockdown efficiency, as one study observed a substantial cell cycle arrest in G2/M (*Li and Wang, 2003*), while another saw no specific arrest in any cell cycle phase (*Monnerat et al., 2013*). In both forms CYC4 transcript levels dip in late G1 before rising again in S phase through to G2M, but only reached an FC >1.5 in BSFs. Interestingly, RNAi against CYC4 in PCF highlighted the cyclin's role in the G1/S transition (*Liu et al., 2013*), again indicating transcript regulation does not predict phenotypic outcome. Finally, a

novel putative cyclin, Tb427_110012500, was detected with CCR transcripts in BSF form only, where transcripts peak between early and late G1. This gene contains a cyclin N-terminal domain, but no functional analysis has been published. Of the remain nine documented cyclins (*Hammarton, 2007*; *Lee and Li, 2021*; *Li, 2012*) in *T. brucei*, none reached significance thresholds in either form.

## Transcript and protein periodicity

Lastly, we compared transcript and protein abundance levels across the cell cycle. In the human cell cycle, just 15% of CCR proteins are encoded by genes which also have CCR transcripts (*Mahdessian et al., 2021*). In this study, we also observed little correlation between transcript and protein regulation during the *T. brucei* cell cycle. Thus, for most genes the cyclic protein abundance patterns are the result of mostly translational, and post-translation processes. Even accounting for experimental differences in approaches, why so many transcripts show cyclic expression patterns without resulting in significant protein changes, especially in the absence of transcriptional control due to polycistronic transcription in *T. brucei* (*Clayton, 2019*; *Clayton, 2016*), remains a puzzling question across eukaryotes. Of those genes that were identified as CCR for both transcript and protein abundance, a relative delay was observed for the majority of genes. A time delay between peak transcript and proteins levels was also observed in human cells (*Mahdessian et al., 2021*). Such a delay may allow *T. brucei* to prepare for the subsequent phase by upregulating transcripts, after which translation can rapidly generate the required proteins. A similar observation can be made during *T. brucei* life cycle progression: stumpy BSFs upregulate hundreds of transcripts related to PCF biology (*Briggs et al., 2021a*; *Kabani et al., 2009*; *Queiroz et al., 2009*; *Silvester et al., 2018*; *Naguleswaran et al., 2018*) in preparation for differentiation, but not all upregulated genes are detectable in proteomic analysis of stumpy forms and instead appear after the rapid development of PCFs once the environmental trigger to differentiate has been received (*Dejung et al., 2016*; *Gunasekera et al., 2012*).

In summary, the experiments discussed here exploit cryopreservation to preserve *T. brucei* for scRNA-seq analysis, an approach that can be likely also be extended to and related parasites, to increase flexibility and feasibility of experimental design. Making use of these data we have generate detailed transcriptome atlases of the BSF and PCF cell cycles, which can be further interrogated by the publicly accessible interactive webtool (https://cellatlas-cxg.mvls.gla.ac.uk/Tbrucei.cellcycle.bsf/ and https://cellatlas-cxg.mvls.gla.ac.uk/Tbrucei.cellcycle.pcf/).

# Materials and methods

## Key resources table

| Reagent type (species) or resource | Designation | Source or reference | Identifiers | Additional information |
|---|---|---|---|---|
| Cell line (*Trypanosoma brucei brucei*) | Lister427 bloodstream forms (BSF) | R.McCulloch stocks (University of Glasgow) | NA | https://tryps.rockefeller.edu/trypsru2_cell_lines.html |
| Cell line (*Trypanosoma brucei brucei*) | Lister427 procyclic form (PCF) | R.McCulloch stocks (University of Glasgow) | NA | https://tryps.rockefeller.edu/trypsru2_cell_lines.html |
| Transfected construct (*Trypanosoma brucei brucei*) | J1339 | *Rojas et al., 2019* Cell 176, 306–317.e16 | NA | NA |
| Sequence-based reagent | Tb927.8.2550 Ntag_F | The study | PCR primers | ATCTGAAGAAAATAATATACAAGAGA CAAGgtataatgcagacctgctgc |
| Sequence-based reagent | Tb927.8.2550 Ntag_R | The study | PCR primers | TTGCTGTGATGGTAAGG TGATGCGGAGCAT actacccgatcctgatccag |
| Sequence-based reagent | Tb927.8.2550 Ntag_sgRNA | The study | PCR primers | gaaattaatacgactcactatagg GCGGGACACGCAACACTACA gttttagagctagaaatagc |
| Sequence-based reagent | Tb927.8.2550_tag_check_F | The study | PCR primers | ATCTGAAGAAAATAATA TACAAGAGACAAG |
| Sequence-based reagent | Tb927.8.2550_tag_check_R | The study | PCR primers | TTGCTGTGATGGTAAG GTGATGCGGAGCAT |
| Sequence-based reagent | Tb927.6.3180_Ctag_F | The study | PCR primers | TTACGAGCGGGACTGCGACGTT CGTGCCTGggttctggtagtggttccgg |

*Continued on next page*

*Continued*

| Reagent type (species) or resource | Designation | Source or reference | Identifiers | Additional information |
|---|---|---|---|---|
| Sequence-based reagent | Tb927.6.3180_Ctag_R | The study | PCR primers | AAGCCTCTGCCGACACGCACATTTC TTCCGccaatttgagagagacctgtgc |
| Sequence-based reagent | Tb927.6.3180_Ctag_sgRNA | The study | PCR primers | gaaattaatacgactcactataggCAATGTG CAGAAGCATAAATgttttagagctagaaatagc |
| Sequence-based reagent | Tb927.6.3180_Ctag_check_F | The study | PCR primers | TTACGAGCGGGACTGCG ACGTTCGTGCCTG |
| Sequence-based reagent | Tb927.6.3180_Ctag_check_R | The study | PCR primers | AAGCCTCTGCCGACAC GCACATTTCTTCCG |
| Sequence-based reagent | Tb927.11.15100_Ntag_F | The study | PCR primers | CTACTTACCCACTGCAGTTTTT TTATTATTgtataatgcagacctgctgc |
| Sequence-based reagent | Tb927.11.15100_Ntag_R | The study | PCR primers | CTACTTACCCACTGCAGTTTTTT TATTATTgtataatgcagacctgctgc |
| Sequence-based reagent | Tb927.11.15100_Ntag_sgRNA | The study | PCR primers | gaaattaatacgactcactataggCGGTAT TACATCAAGTAAAGgttttagagctagaaatagc |
| Sequence-based reagent | Tb927.11.15100_Ntag_check_F | The study | PCR primers | CTACTTACCCACTGCA GTTTTTTTATTATT |
| Sequence-based reagent | Tb927.11.15100_Ntag_check_R | The study | PCR primers | ATCGGCAAAGTTCTTGTG GACAACGGCCAT |
| Commercial assay or kit | Chromium Single Cell 3′ v3.1 | 10× Genomics | SCR_019326 | NA |
| Software, algorithm | R | https://www.r-project.org/ | RRID: SCR_001905 | NA |
| Software, algorithm | GraphPad Prism | https://www.graphpad.com | RRID: SCR_002798 | NA |
| Software, algorithm | Rstudio | https://rstudio.com/ | RRID: SCR_000432 | NA |
| Software, algorithm | Cellranger version 7 | 10× Genomics | N/A | NA |
| Software, algorithm | Seurat version 4.1.0 | *Hao et al., 2021* | RRID: SCR_007322 | NA |
| Software, algorithm | Complete scRNA-seq analysis code | This paper, Zenodo | DOI: 10.5281/zenodo.7508131 | NA |
| Other | TritrypDB database | http://tritrypdb.org/tritrypdb/ | N/A | TritrypDB database for searching genome |
| Other | 10× Genomics Chromium Plus Genetic Analyzer | 10× Genomics | SCR_019326 | 10× controller for cell sorting into droplets |
| Other | SDM-79 Medium | Life Technologies | Cat# RR110008P1 | Medium for PCF culture |
| Other | HMI-9 Medium | Life Technologies | Cat# 074-90915 | Medium for BSF culture |

## Cell lines

*Trypanosoma brucei brucei* Lister 427 BSF and PCF cell lines were sourced from the R. McCulloch lab (University of Glasgow). scRNA-seq data made from these cell lines showed no evidence of contamination with sequence reads from Mycoplasma and confirmed the identity of these lines as *T. brucei* Lister427.

## *T. brucei* culture

For scRNA-seq experiments, BSF Lister 427 were cultured in HMI-9 (*Hirumi and Hirumi, 1989*) with 20% foetal calf serum (FCS) at 37°C with 5% $CO_2$. PCF Lister 427 were cultured in SDM-79 (*Brun, 1979*) supplemented with 10% FCS and 0.2% hemin, at 27°C in sealed flasks without $CO_2$. A haemocytometer was used for all cell density and motility counts. For mNeonGreen tagging experiments Lister 427 BSF expressing Cas9 were used (gifted, R. McCulloch). These had been transfected with J1339 plasmid (*Rojas et al., 2019*), which allows constitutive expression of Cas9.

For cryopreservation of both PCF and BSF, fresh 2× freezing media with FCS and 20% glycerol was used for each sample. Cell density was adjusted to $2 \times 10^6$/ml before parasite culture and 2× freezing media were mixed 1:1 by slow addition of freezing media to culture and gentle resuspension. Cells

were aliquoted into 1 ml cryopreservation tubes, wrapped in cotton wall to prevent rapid cooling and incubated at −80°C for 24 hr. Tubes were then moved to $LN_2$ storage. 1 ml samples were thawed and used immediately for scRNA-seq library preparation. Tubes were placed at room temperature (RT) for 5–10 min before incubating at 37°C (BSF) or 27°C (PCF), until a small ice crystal was left in the tube. Cells were moved to RT until completely defrosted then pipetted with wide-bore pipette tips into 50 ml falcons. 1 ml of pre-warmed media with FCS (37 or 27°C as appropriate) was added drop-wise to falcon with and swirled gently. A further 1 ml of media was used to rinse the cryotube with a wide-bore tip and added drop-wise to falcon. Increasing volumes of pre-warmed media was added to the cells (3, 6, and 12 ml) drop-wise, with at least 1 min pause between each addition. Cells were pelleted by centrifugation at $400 \times g$, for 10 min at RT, and the supernatant was poured off. 10 ml of media was then added dropwise to wash cells. Cells were centrifuged again and supernatant poured off, before resuspending in 1 ml of 1× phosphate-buffered saline (PBS) supplemented with 1% D-glucose (PSG) and 0.04% bovine serum albumin (BSA), by gentle pipetting. Cells were strained through a 40-μm filter into a 1.5 ml Eppendorf. Cells were centrifuged at $400 \times g$ for 10 min at RT and supernatant was removed with a pipette. Cells were suspended in 150 μL of PSG + 0.04% BSA. Sample was diluted 1:1 and in PSG + 0.04% BSA and loaded to haemocytometer to determine cell concentration. Cell concentration was adjusted to 1000 cells/μl and stored on ice.

## scRNA-seq sample preparation of fresh in vitro cultured *T. brucei*

For both BSF and PCF *T. brucei*, $1 \times 10^6$ cells were transferred to a falcon tube and were centrifuged at $400 \times g$ for 10 min at RT. The supernatant was poured off and pre-warmed media added drop-wise to the sample to wash cells. Cells were centrifuged again and supernatant poured off, before resuspending in 1 ml of PSG + 0.04% BSA, by gentle pipetting with wide-bore pipette tips. Cells were strained through a 40-μm filter into a 1.5-ml Eppendorf before centrifuging again at $400 \times g$ for 10 min at RT removing the supernatant with a pipette. Cells were suspended in 150 μl of PSG + 0.04% BSA and concentration adjusted to 1000 cells/μl before storing on ice.

## Flow cytometry analysis

For PCFs, parasites were wash in 1 ml of wash buffer (1× PBS with 5 mM of ethylenediaminetetraacetic acid (EDTA) and 1% fetal bovine serum) before fixing in 70% cold methanol (in wash buffer) over night. PCFs were washed again before resuspending in wash buffer supplemented with 10 μg/ml of propidium iodide (PI) and 10 μg/ml of RNaseA and incubating at 37°C for 45 min. BSFs were instead fixed with 1% formaldehyde in wash buffer at room temperature for 10 min, before washing and permeablising with 0.01% Triton X-100 in wash buffer at room temperature for 30 min. BSFs were then washed and stained with 10 μg/ml of PI and 100 μg/ml of RNaseA as for PCFs. Samples were filtered with a pluriStrainer Mini (40 um) before 10,000 events were captured with BD Celesta to measure PI-stained DNA content. For frozen samples, flow cytometry was performed with samples at the point of freezing for future scRNA-seq using the same method.

## Chromium (10× Genomics) library preparation and Illumina sequencing

As BSF and PCF are easily identified by known transcriptional differences (*Siegel et al., 2010*; *Kabani et al., 2009*; *Queiroz et al., 2009*; *Jensen et al., 2014*; *Naguleswaran et al., 2018*; *Vasquez et al., 2014*), the two forms were multiplexed. Fresh BSF and PCF were combined in approximate equal ratio into sample 1, and cryopreserved BSF and PCF into sample 2. 14,000 cells of each sample were loaded onto the Chromium Control and library preparation was performed with the Chromium Single Cell 3′ chemistry version 3.1 kits. (*L. major* parasites were additionally multiplexed with each sample as performed previously with kinetoplastids (*Briggs et al., 2021a*), but are not analysed here.) Libraries were sequences with Illumina NextSeq 2000, to generate 28 × 130 bp paired reads to a depth of 46,561 and 43,332 mean reads per cell for samples 1 and 2, respectively. Library preparation and sequencing were performed by Glasgow Polyomics.

## Data mapping and count matrix generation

To improve the proportion of mapped reads attributed to a feature for transcript counting, the UTR annotation of the Lister 427 2018 reference genome (*Müller et al., 2018*) were extended. 2500 bp were added to the end each annotated coding region of the gtf file (unless the annotation overlapped

with the next genomic feature in which case the UTR was extended to the base before the next feature). The same approach was used to edit the *L. major* Friedlin reference genome annotation (*Ivens et al., 2005*). Reads were mapped to both the *T. brucei* WT427 2018 and *L. major* Friedlin references and counts matrix generated with Cell ranger v 7. *L. major* transcriptomes and those of multiplets containing transcripts from both species were removed from analysis. The resulting count matrices and samples summaries are available in *Supplementary file 1* and at Zenodo (10.5281/zenodo.7508131).

## Sample de-multiplexing and QC filtering

To de-multiplex the PCF and BSF transcriptomes, a set of high confidence marker genes was defined from published bulk-RNA-seq studies where two replicates are available for DE analysis. DE analysis was performed using TriTrypDB (*Amos et al., 2022*) which implements DESeq2 (*Love et al., 2014*) to compare datasets. DE between Lister 427 PCFs and Lister427 monomorphic BSFs (*Jensen et al., 2014*), and slender pleomorphic BSF EATRO 1125 (clone AnTat 1.1) and experimentally derived early PCFs (*Naguleswaran et al., 2018*), identified 238 BSF and 221 PCF marker genes (FC >2, p-value <0.05, *Supplementary file 1*). As PCFs and BSFs were expected to be present at around a 1:1 ratio, marker genes detected in 20–70% of the cells were selected as markers (*Supplementary file 1*). This gave 157 and 50 high confidence marker genes for PCFs and BSFs, respectively. scGate (*Andreatta et al., 2022*) was used to gate BSF, PCF, multiplets containing a mix of each life cycle form using marker genes and transcriptomes not enriched for either form (*Supplementary file 1*). Once, demultiplexed into each sample (BSF fresh, BSF frozen, PCF fresh, and PCF frozen) cells were filtered for homogenous multiples with higher-than-average UMI and feature counts, and poor-quality transcriptomes with low UMI and feature counts (*Figure 1A, B*). Finally, cells expressing higher than average mitochondrial transcripts encoded on the kDNA maxi circle were removed, as these were likely generated from lysing cells (*Figure 1C*). Full code is available for all steps at Zenodo (10.5281/zenodo.7508131).

## Live vs cryopreserved *T. brucei* DE analysis

The AverageExpression function from Seruat v4.1.0 (*Hao et al., 2021*) was used to average expression of each gene across cells for each sample. Fold changed was calculated as average expression for frozen over fresh samples for each life cycle form separately. Genes with average expression <0.05 counts in each fresh or frozen were excluded from fold change analysis. For PCA analysis, data were 'pseudobulked' by summing counts across all cells for each gene, per condition. DESeq2 v1.32.0 was used to log2 scale counts and generate PCA plot. DE analysis between individual transcriptomes of each condition was performed with Seurat v4 (*Hao et al., 2021*) function FindAllMarkers using MAST test (*Finak et al., 2015*). Only genes detected in 25% of cells in tested condition and with FC >1.5 between fresh and frozen were considered.

## Data integration and dimensional reduction

Each sample was normalised and log2 transformed using Scran v1.22.1 (*Lun et al., 2016*), as described previously (*Briggs et al., 2021a*). The top 3000 variable genes were identified in each sample using two independent methods (Scran, which using log2 counts and Seurat applied to raw counts), and results compared to select common variable genes. 1939 genes were identified for BSF fresh, 2063 for BSF frozen, 2000 for PCF fresh, and 1924 for PCF frozen (*Supplementary file 1*). For data integration, variable genes for fresh and frozen samples were compared and selected using SelectIntegrationFeatures before filtering for only those with standardised variance over 1 in both conditions. BSF and PCF samples were considered separately, identifying 1652 and 1738 variable genes for integration, respectively (*Supplementary file 1*). Integration was performed with fast mutual nearest neighbours (FastMNN), which performs batch correction by finding MNN pairs of cells between conditions with mutually similar gene expression and calculating correction between these pairs. MNN does not assume equal population composition between sample and only performs correction between the overlapping subsets of cells (*Haghverdi et al., 2018*). FastMNN first performs a PCA across all cells and finds MNN between cells in this deduced dimensional space to increase speed and remove noise. The default of 50 dimensions was used to integrate fresh and frozen samples for BSF and PCF independently, and nearest-neighbours were identified for 5% of cells in each case. Integrated cells

were visualised using UMAP (*McInnes et al., 2018*) applied to the first 30 dimensions calculated by FastMNN, implemented by the Seurat package.

## Cell cycle phase labelling

Cell cycle phases were inferred using marker gene identified with bulk RNA-seq previously (*Archer et al., 2011*). Syntenic orthologs for each phase marker (originally identified in the TRUE927 genome) were found of Lister 427 2018 reference genome via TritrypDB (*Amos et al., 2022*), and those detected in at least 10% of transcriptomes were selected for PCF and BSF integrated datasets independently. An 'expression score' for each phase was found of each using MetaFeature function from Seurat using markers. The ratio of a cell's expression score over the mean expression scores across cells was calculated for each phase. The phase with the highest ratio was assigned to each cell. If a cell has an expression score <1 for all phase (i.e. no enrichment over the average phase score), the cell was assigned 'unlabelled'.

## Pseudotime inference and DE analysis

For pseudotime inference the autoencoder approach from Cyclum (*Liang et al., 2020*) was used for BSF and PCF separately. Counts for the same variable genes used for integration steps, described above, were first scaled before the model was trained using 25% of total cells and default parameters. The model was then applied to the whole dataset to infer pseudotime values for each cell. To allow clear visualisation and comparison between PCF and BSF cell cycles, pseudotime was scaled between 0 and 1 for each form and in the case of PCFs, pseudotime was shifted to set 0 to be at approximated early G1.

PseudotimeDE (*Song and Li, 2021*) was used for DE analysis over pseudotime. This package calculates accurate p-values by accounting for uncertainty in the pseudotime inference, and shows greater power and lower FDRs than similar packages (*Song and Li, 2021*). To calculate pseudotime uncertainty, the same Cyclum training model was applied to 100 subsets of the data, each containing 80% of the cells selected at random. Genes that were detected in at least 10% of the cells were assessed for DE over pseudotime using a negative binomial generalised additive model (NB-GAM) and default settings. The empirical p-values calculated by PseudotimeDE (which take into account pseudotime uncertainty) was adjusted using the Benjamini and Hochberg method to find the FDR (*Benjamini and Hochberg, 1995*). PseudotimeDE was applied to the normalised log-transformed counts, to account for overall increase in RNA across the cell cycle (*Figure 2—figure supplement 1*).

For calculating FC in gene expression over pseudotime, the smoothed expression of each gene was predicted from the GAM fitted by PseudotimeDE. The ratio of the maximum value in this prediction over the minimum value was calculated as the FC in the average expression over pseudotime. Genes were considered CCR if adjusted p-value was below 0.01, and FC was over 1.5, based on the detection of known CCR genes (*Archer et al., 2011*; *Figure 2—figure supplement 2*). Predicted models were also used when plotting smoothed expression. All GO term enrichment analysis was performed using the TriTrypDB resource (*Amos et al., 2022*).

## Dataset comparison

For comparison with proteomics (*Crozier et al., 2018*; *Benz and Urbaniak, 2019*), bulk transcriptomics (*Archer et al., 2011*), and genome-scale cell cycle defect RNAi screen (*Marques et al., 2022*), all of which used the TRUE927 reference genome (*Berriman et al., 2005*), syntenic orthologs were identified in the Lister 427 2018 reference (*Müller et al., 2018*) using TriTrypDB implementation of OrthoMCL (*Li et al., 2003*). For each study, CCR genes were retained as those selected by original authors.

## Gene conservation analysis

Orthogroups were identified for each CCR gene common to both BSF and PCF cell cycles, and orthologous protein sequences across 44 kinetoplastida proteomes were extracted from previous analysis (*Oldrieve et al., 2022*). A distance matrix was created from orthologous protein sequences with ClutalOmega (*Sievers et al., 2011*). Using the distance matrix, FastME (*Lefort et al., 2015*) was used to calculate the tree length for each orthogroup which contained four or more protein sequences.

## Expression profiling of mNeonGreen tagged proteins

CRISPR/Cas9 editing was used to added epitope tags to three genes in BSF WT427/Ca9 (Tb427_080028700, Tb427_060036900, and Tb427_110169500). Gene-specific primers were used to amplify the donor fragment containing mNeonGreen and G418 resistance gene from a pPOTv7 plasmid as previously designed (*Beneke et al., 2017*; *Beneke and Gluenz, 1971*). Primers were designed using the TREU927 syntenic homolog and the LeishGEdit.net resource (*Beneke et al., 2017*; *Beneke and Gluenz, 1971*), and are provided in the Key resources table. 30 ng circular plasmid, 0.2 mM dNTPs, 2 µM each of gene-specific forward and reverse primers and 1 unit Phusion polymerase (New England Biolabs) were mixed in 1× HF Phusion buffer and 3% (vol/vol) dimethyl sulfoxide (DMSO), up 50 µl total volume with $H_2O$. The PCR was run as follows: 5 min at 98°C, 40 cycles of 98°C for 30 s, 65°C for 30 s , and 72°C for 2 min 15 s, followed by a final extension at 72°C for 7 min. To amplify the sgRNA, 2 µM of gene-specific forward primer, 2 µM of the generic G00 primer (*Beneke et al., 2017*), 0.2 µM of dNTPs, 1 unit of Phusion Polymerase were mixed with 1× HF Phusion buffer (NEB), and made up to 50 µl total volume with $H_2O$. The PCR was run as follows: 98°C for 30 s, followed 35 cycles of 98°C 10 s, 60°C for 30 s, and 72°C for 15 s. 2 µl of each product was run on 1% agarose gel to confirm expected size and the products were both ethanol precipitated and eluted into 5 µl of $H_2O$. 1 × 10$^7$ WT427/Cas9 BSFs were transfected in 100 µl of transfection buffer (90 mM $NaH_2PO_4$, 5 mM KCl, 150 µM $CaCl_2$ and 500 mM HEPES (sodium 2-[4-(2-hydroxyethyl)piperazin-1-yl]ethane-1-sulfonate, pH 7.3) plus the 5 µl donor and 5 µl sgRNA, using the Nucleofector 2b Device (Lonaz) using program X-100. Parasites were serially diluted and aliquoted into 24-well plated. G418 selection was added after 16–24 hr at final concentration of 2 µg/ml and clones were recovered after 5–7 days. To confirm tag integration with PCR, genomic DNA was extracted from WT427 cas9 BSFs and three clonal derivatives for each gene using the DNeasy Blood and Tissue extraction kit (QIAGEN). 5 µM each of gene-specific forward and reverse primers and 30 ng of gDNA was mixed with 0.4 µl Phire Hot Start II polymerase, 1× Phire Reaction Buffer, 0.2 mM dNTPs and up to 20 µl $H_2O$. The two-step PCR was run as follows: 30 s at 98°C, 30 cycles of 98°C for 5 s, and 72°C for 1 min, followed by a final extension at 72°C for 1 min. For fluorescence and flow cytometry assays, cells were harvested by centrifugation at 400 × *g* for 10 min, washed in 1× PBS and fixed in 1% formaldehyde for 10 min at room temperature. Cells were pelleted and washed again in 1× PBS to remove formaldehyde. For microscopy, cells were attached to a poly-L-lysine treated slide before 5 µl of Fluoromount G with DAPI (Cambridge Bioscience, Southern Biotech) was added and coverslip applied. For flow cytometry, formaldehyde fixed cells were resuspended in 1× PBS supplemented with 5 mM EDTA and 0.1 µg/ml DAPI and incubated on ice for 30 min. DAPI and mNeonGreen fluorescence were detected for 10,000 events per sample.

## Acknowledgements

We thank J Galbraith and P Herzyk (Glasgow Polyomics, University of Glasgow) for their guidance, library preparation, sequencing, and data handling. We also thank R McCulloch (University of Glasgow) for provision of cell lines and manuscript comments, and E Agboraw (University of Glasgow) for uploading data to the webtool.

## Additional information

### Funding

| Funder | Grant reference number | Author |
| --- | --- | --- |
| Wellcome Trust | 218648/Z/19/Z | Emma M Briggs |
| Wellcome Trust | 104111/Z/14/ZR | Thomas D Otto |
| Wellcome Trust | 221717/Z/20/Z | Keith R Matthews |
| Wellcome Trust | 220058/Z/19/Z | Guy R Oldrieve Keith R Matthews |

| Funder | Grant reference number | Author |
|---|---|---|
| Biotechnology and Biological Sciences Research Council | BB/R017166/1 | Catarina A Marques |
| Biotechnology and Biological Sciences Research Council | BB/W001101/1 | Catarina A Marques |

The funders had no role in study design, data collection and interpretation, or the decision to submit the work for publication. For the purpose of Open Access, the authors have applied a CC BY public copyright license to any Author Accepted Manuscript version arising from this submission.

## Author contributions

Emma M Briggs, Conceptualization, Resources, Data curation, Formal analysis, Supervision, Funding acquisition, Validation, Investigation, Visualization, Methodology, Writing - original draft, Project administration, Writing - review and editing; Catarina A Marques, Data curation, Formal analysis, Methodology, Writing - review and editing; Guy R Oldrieve, Conceptualization, Formal analysis, Methodology, Writing - review and editing; Jihua Hu, Conceptualization, Methodology; Thomas D Otto, Conceptualization, Resources, Visualization, Methodology, Writing - review and editing; Keith R Matthews, Conceptualization, Supervision, Funding acquisition, Methodology, Writing - review and editing

## Author ORCIDs

Emma M Briggs (ID) http://orcid.org/0000-0002-6740-8882
Catarina A Marques (ID) http://orcid.org/0000-0003-1324-5448
Thomas D Otto (ID) http://orcid.org/0000-0002-1246-7404
Keith R Matthews (ID) http://orcid.org/0000-0003-0309-9184

## Decision letter and Author response

Decision letter https://doi.org/10.7554/eLife.86325.sa1
Author response https://doi.org/10.7554/eLife.86325.sa2

## Additional files

### Supplementary files

• Supplementary file 1. Sample processing. (Tab 1) Genes used for BSF and PCF gating. (Tab 2) Results of BSF and PCF gating. (Tab 3) scRNA-seq sample summary. (Tab 4) Variable genes per sample. (Tab 5) Cell cycle phase markers used for labelling.

• Supplementary file 2. Fresh vs frozen DE analysis. (Tab 1) BSF average expression comparison. (Tab 2) PCF average expression comparison. (Tab 3) BSF MAST DE test. (Tab 4) PCF MAST DE test.

• Supplementary file 3. Early G1 vs unlabelled cells comparison. (Tab 1) PCF DE. (Tab 2) BSF DE.

• Supplementary file 4. PCF cell cycle DE analysis. (Tab 1) PseudotimeDE analysis. (Tab 2) PseudotimeDE results for CCR genes only. (Tab 3) Average expression of each CCR per phase. (Tab 4) Dataset comparison of CCR genes. (Tab 5) Comparison with Crozier proteomics study. (Tab 6) Comparison with Benz proteomics study.

• Supplementary file 5. BSF cell cycle DE analysis. (Tab 1) PseudotimeDE analysis. (Tab 2) PseudotimeDE results for CCR genes only. (Tab 3) Average expression of each CCR per phase. (Tab 4) Comparison with Crozier proteomics study. (Tab 5) Comparison with Benz proteomics study.

• Supplementary file 6. Common CCR gene analysis. (Tab 1) CCR comparison between BSF and PCF. (Tab 2) Comparison of peak expression between BSF and PCF. (Tab 3) Ortholog groups and conservation of common CCR genes. (Tab 4) Comparison of CCR genes and cell cycle defects. (Tab 5) Early G1 phase associated GO terms. (Tab 6) Late G1 phase associated GO terms. (Tab 7) S phase associated GO terms. (Tab 8) G2M phase associated GO terms. (Tab 9) Plotted GO terms (Tab 10) CCR RNA-binding proteins.

• Supplementary file 7. Unique CCR gene analysis. (Tab 1) GO terms associated with CCR unique to PCF. (Tab 2) GO terms associated with CCR unique to BSF.

• Supplementary file 8. Primer sequences.

• MDAR checklist

## Data availability

The transcriptome data generated in this study have been deposited in the EuropeanNucleotide Archive with project accession number PRJEB58781. The processed transcript count data and cell metadata generated in this study as well as all code and necessary intermediate files are available at Zenodo (10.5281/zenodo.7508131). BSF and PCF cell cycle transcriptomes can also explored using the interactive cell atlas (https://cellatlas-cxg.mvls.gla.ac.uk/Tbrucei.cellcycle.bsf/ and https://cellat-las-cxg.mvls.gla.ac.uk/Tbrucei.cellcycle.pcf/).

The following datasets were generated:

| Author(s) | Year | Dataset title | Dataset URL | Database and Identifier |
|---|---|---|---|---|
| Briggs EM, Marques CA, Oldrieve GR, Hu J, Otto TD, Matthews KR | 2023 | Profiling the bloodstream form and procyclic form *Trypanosoma brucei* cell cycle using single cell transcriptomics | https://www.ebi.ac.uk/ena/browser/view/PRJEB58781 | ENA, PRJEB58781 |
| Briggs EM | 2023 | Single cell transcriptomic analysis of the bloodstream form and procyclic form Trypanosoma brucei cell cycle | https://doi.org/10.5281/zenodo.7508131 | Zenodo, 10.5281/zenodo.7508131 |

The following previously published datasets were used:

| Author(s) | Year | Dataset title | Dataset URL | Database and Identifier |
|---|---|---|---|---|
| Benz C, Urbaniak MD | 2019 | Organising the cell cycle in the absence transcriptional control: Dynamic phosphorylation co-ordinates the *Trypanosoma brucei* cell cycle post-transcriptionally | http://proteomecentral.proteomexchange.org/cgi/GetDataset?ID=PXD013488 | ProteomeXchange, PXD013488 |
| Crozier TWM, Tinti M, Wheeler RJ, Ly T, Ferguson MAJ, Lamond AI | 2018 | Proteomic analysis of the cell cycle of procylic form *Trypanosoma brucei* | http://proteomecentral.proteomexchange.org/cgi/GetDataset?ID=PXD008741 | ProteomeXchange, PXD008741 |

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
