## [Editor Report]

This important study maps changes in transcript levels over the cell cycle of two major developmental stages of the parasitic protist, Trypanosoma brucei. Single-cell RNA-seq on asynchronously replicating insect and mammlain-infective parasite stages identified over 1500 transcripts that are cell cycle regulated, significantly expanding the number of genes and cellular processes linked to cell cycle progression. Significantly, only some of these transcript levels are reflected in changes in corresponding protein levels, underlining the importance of both pre- and post-transcriptional regulatory processes in these divergent eukaryotes.

---

## [Decision Letter]

**Decision letter after peer review:**

Thank you for submitting your article "Profiling the bloodstream form and procyclic form Trypanosoma brucei cell cycle using single cell transcriptomics" for consideration by *eLife*. Your article has been reviewed by 3 peer reviewers, including Malcolm J McConville as the Reviewing Editor and Reviewer #1, and the evaluation has been overseen by Dominique Soldati-Favre as the Senior Editor. The following individual involved in the review of your submission has agreed to reveal their identity: Marcelo S. da Silva (Reviewer #3).

Essential revisions:

Please address all of the methodology-related comments and, as far as possible, some of the more general points raised in the three reviews.

*Reviewer #1 (Recommendations for the authors):*

One of the most striking findings of this study is the lack of concordance between levels of the majority of transcripts and corresponding proteins in PCF, as well as some of the transcripts tested in BSF. While defining the post-transcriptional regulatory mechanisms in trypanosomatids is complex and beyond this study, it would be of interest to document the extent to which transcripts for different RNA binding proteins are cell cycle regulated, given the documented role of some of these proteins in regulating mRNA stability and protein translation.

The authors note that some of the T. brucei CCR genes may be linked to the cycling of metabolic processes, as is well established in other eukaryotes. It would be interesting to know whether T. brucei CCR genes are potentially linked to other cycles, most notably the putative circadian rhythm that has recently been reported to involve 10% of T. brucei PCF/BSF transcripts.

*Reviewer #2 (Recommendations for the authors):*

Congratulations to this excellent research team for the in-depth analysis of trypanosome developmental processes. Please consider the suggestions below for enriching your analysis and readership.

Figure 1 The authors should consider, in either the methods section or in the supplement, stating which cell-cycle phase markers were used in the determination of cell-cycle phases. While they refer to a publication, this information is of fundamental nature and pertinent for the readers to understand this analysis.

Figure 2D. How was data on the proportion of cells with DNA content assessed by flow cytometry performed?

Figure 2G. The authors state "CCR genes defined by scRNA-seq above were compared with CCR proteins. Transcript levels for those identified in the two proteomics studies are shown in Figure 2G). Could the authors indicate that these data were generated from cultured procyclic cells (line 238)? Could the authors comment on the abundance of these identified particular proteins in their scRNA data? Were proteins for the 586 scRNA-seq CCR transcripts not detected as they represent low abundant transcripts? Some discussion would be helpful.

The web interphase is very helpful and will be a good resource for the community.

Discussion:

Identification of cell-cycle regulators, such as RNA binding proteins, would be an important discovery. Could the authors describe their findings as such.

Following this descriptive analysis, a discussion on the downstream functional investigations would be important for the readers and the field, particularly with translational implications.

Given that the data presented are generated from in vitro cell lines, what are some of the limitations these findings may reflect?

Collective evidence indicates a delay between transcript abundance and protein level data. What are some of the conceptual implications of this observation for functional studies in organisms with significant post-transcriptional regulation?

*Reviewer #3 (Recommendations for the authors):*

This study is sound and provides a detailed atlas of the cell cycle-regulated transcriptomes of both replicative forms of T. brucei (PCF and BSF). My main comments are relatively minor and contain only a few inquiries.

Regarding cryopreserved samples, it was not clear whether the authors started the scRNAseq library preparation immediately after thawing frozen samples, or after 4 days (as indicated in figure S2). If samples started to be processed immediately after thawing, I suggest a flow cytometric DNA content analysis (as was done in Figure S2). This is important to confirm the absence of possible cell cycle arrests. However, if the samples were processed after 4 days of recovery, I would like to know how the authors explain the sharp G1 phase cell cycle arrest in BSF frozen cells after 96 h (Figure S2C). This may (very likely) be introducing a bias in the results. I believe the increased proportion of cells in the G1 phase in Figure 3C may be related to this initial cell cycle arrest.

---

## [Author Response]

Essential revisions:Please address all of the methodology-related comments and, as far as possible, some of the more general points raised in the three reviews.Reviewer #1 (Recommendations for the authors):One of the most striking findings of this study is the lack of concordance between levels of the majority of transcripts and corresponding proteins in PCF, as well as some of the transcripts tested in BSF. While defining the post-transcriptional regulatory mechanisms in trypanosomatids is complex and beyond this study, it would be of interest to document the extent to which transcripts for different RNA binding proteins are cell cycle regulated, given the documented role of some of these proteins in regulating mRNA stability and protein translation.

We have now mined the cell cycle regulated (CCR) genes for those with documented RNA-binding domains which were highlighted in this study as CCR themselves, as well as known targets of previously characterised RNA binding proteins, ZC3H20/21 and RBP10. These additions are in the Results sections (lines 358-365 and lines 443-464). Supplementary files 4, 5 and 6 have been updated to include the pfam domains for each CCR gene in both PCF and BSFs highlight the common and unique genes with RNA binding domains, and to note the targets of RBP10 and ZC3H20/21.

The authors note that some of the T. brucei CCR genes may be linked to the cycling of metabolic processes, as is well established in other eukaryotes. It would be interesting to know whether T. brucei CCR genes are potentially linked to other cycles, most notably the putative circadian rhythm that has recently been reported to involve 10% of T. brucei PCF/BSF transcripts.

We have now compared the CCR and circadian rhythm regulated transcripts identified by Rijo-Ferreira et al. (2017, Nature Microbiology). Of the BSF CCR transcripts, 243 were also found to oscillate under constant conditions by Rijo-Ferreria et al., equating to 13% of the CCR transcriptome and not significantly different to the percentage of all genes which were found to oscillate in that study (~15% of the *T. brucei* genome). The CCR transcriptome is therefore not enriched for circadian regulated transcripts. These 243 genes however represent 22.3% of the BSF circadian oscillating transcripts found in the previous study. This list of genes is enriched for those associated with GO terms including “glucose metab­olic process”, “microtubule-based movement” and “carboxylic acid metabolic process”. We therefore find that perhaps the cell cycle regulation has some overlap in the circadian rhythms documented. However, as most of these genes are linked to metabolism, this more likely reflects that both processes can influence levels of these mRNAs, and less likely that cell cycle dynamics are changing during the circadian rhythms, in agreement with the author’s original findings. As validating this conclusion is beyond the scope of this study, we have not added this observation to the manuscript.

Reviewer #2 (Recommendations for the authors):Congratulations to this excellent research team for the in-depth analysis of trypanosome developmental processes. Please consider the suggestions below for enriching your analysis and readership.

We would like to thank the reviewer for their helpful suggestions and positive feedback. We have made the following changes to the manuscript, as detailed below.

Figure 1 The authors should consider, in either the methods section or in the supplement, stating which cell-cycle phase markers were used in the determination of cell-cycle phases. While they refer to a publication, this information is of fundamental nature and pertinent for the readers to understand this analysis.

These marker genes are now listed in supplementary file 1 (new tab 5), as well as the zenodo web page, and are referred to in the text (line 159).

Figure 2D. How was data on the proportion of cells with DNA content assessed by flow cytometry performed?

We apologise that details of these experiments were not included; a section has now been added to the manuscript methods section (lines 729-740).

Figure 2G. The authors state "CCR genes defined by scRNA-seq above were compared with CCR proteins. Transcript levels for those identified in the two proteomics studies are shown in Figure 2G).

This has been added (line 199).

Could the authors indicate that these data were generated from cultured procyclic cells (line 238)? Could the authors comment on the abundance of these identified particular proteins in their scRNA data? Were proteins for the 586 scRNA-seq CCR transcripts not detected as they represent low abundant transcripts? Some discussion would be helpful.

Thank you for the interesting suggestion, upon investigation we find the CCR transcripts without detectable protein in previous studies at no lower than average in our PCF data set. The data is available in the supplementary data, which we have plotted below for the reviewer. We have highlighted this now in the text and pointed to the supplementary data (lines 214-215).

**Author response image 1. sa2fig1:** 

The web interphase is very helpful and will be a good resource for the community.Discussion:Identification of cell-cycle regulators, such as RNA binding proteins, would be an important discovery. Could the authors describe their findings as such.Following this descriptive analysis, a discussion on the downstream functional investigations would be important for the readers and the field, particularly with translational implications.

RNA binding proteins were also highlighted as interesting by reviewer one and so we have added these findings to the manuscript and data files. Please see our response to reviewer one above for details.

Given that the data presented are generated from in vitro cell lines, what are some of the limitations these findings may reflect?

As our study focuses on the active replication cycle, this is most directly addressed with in vitro culture to avoid the confounding influences of differentiation (and associated changes to the cell cycle) concurrent with the in vivo life cycle transitions. Comparison with in vivo data in the future will be interesting and may reveal cell cycle dynamic difference between various host environments, but are beyond the scope of this study.

Collective evidence indicates a delay between transcript abundance and protein level data. What are some of the conceptual implications of this observation for functional studies in organisms with significant post-transcriptional regulation?

The study does re-emphasise that transcript levels and protein levels can often poorly correlate. Inevitably, perturbation studies based on gene silencing/RNAi of transcripts may therefore not identify anticipated functional read-outs for a silenced transcript if its protein is not ordinarily present.

Reviewer #3 (Recommendations for the authors):This study is sound and provides a detailed atlas of the cell cycle-regulated transcriptomes of both replicative forms of T. brucei (PCF and BSF). My main comments are relatively minor and contain only a few inquiries.

Thank you for reviewing our study, we have addressed the comments below.

Regarding cryopreserved samples, it was not clear whether the authors started the scRNAseq library preparation immediately after thawing frozen samples, or after 4 days (as indicated in figure S2). If samples started to be processed immediately after thawing, I suggest a flow cytometric DNA content analysis (as was done in Figure S2). This is important to confirm the absence of possible cell cycle arrests. However, if the samples were processed after 4 days of recovery, I would like to know how the authors explain the sharp G1 phase cell cycle arrest in BSF frozen cells after 96 h (Figure S2C). This may (very likely) be introducing a bias in the results. I believe the increased proportion of cells in the G1 phase in Figure 3C may be related to this initial cell cycle arrest.

Frozen scRNA-seq samples were processed immediately after thawing. This was done to avoid inevitable transcriptome changes and selection that would take place if parasites were allowed to recover in culture after thawing. The growth curve in Figure S2 indicates the growth prior to freezing. The flow cytometry was performed on the samples at the point of freezing. We believed the slight difference in cell cycle phase proportions between fresh and frozen BSFs (reflected in both transcriptome and DNA content analysis) are due to samples being in slightly different growth phases, the former being processed at slightly higher cell density than the latter.

We have now edited the results (line 164-165), methods (line 699) and figure legend (lines 1385 – 1389) text to improve the clarity of this.